# Spatial regulation by multiple *Gremlin1* enhancers provides digit development with *cis*-regulatory robustness and evolutionary plasticity

Jonas Malkmus[1,9], Laurène Ramos Martins[1,9], Shalu Jhanwar [1,2], Bonnie Kircher[3], Victorio Palacio[1], Rushikesh Sheth[1], Francisca Leal[3], Amandine Duchesne[4], Javier Lopez-Rios [5], Kevin A. Peterson [6], Robert Reinhardt[1], Koh Onimaru [7], Martin J. Cohn [3,8], Aimée Zuniga [1✉] & Rolf Zeller [1✉]

Precise *cis*-regulatory control of gene expression is essential for normal embryogenesis and tissue development. The BMP antagonist *Gremlin1* (*Grem1*) is a key node in the signalling system that coordinately controls limb bud development. Here, we use mouse reverse genetics to identify the enhancers in the *Grem1* genomic landscape and the underlying *cis*-regulatory logics that orchestrate the spatio-temporal *Grem1* expression dynamics during limb bud development. We establish that transcript levels are controlled in an additive manner while spatial regulation requires synergistic interactions among multiple enhancers. Disrupting these interactions shows that altered spatial regulation rather than reduced *Grem1* transcript levels prefigures digit fusions and loss. Two of the enhancers are evolutionary ancient and highly conserved from basal fishes to mammals. Analysing these enhancers from different species reveal the substantial spatial plasticity in *Grem1* regulation in tetrapods and basal fishes, which provides insights into the fin-to-limb transition and evolutionary diversification of pentadactyl limbs.

[1] Developmental Genetics, Department of Biomedicine, University of Basel, Basel, Switzerland. [2] Swiss Institute for Bioinformatics, University of Basel, Basel, Switzerland. [3] Department of Biology, University of Florida, Gainesville, FL, USA. [4] Université Paris-Saclay, INRAE, AgroParisTech, GABI, Jouy-en-Josas, France. [5] Development and Evolution, Centro Andaluz de Biología del Desarrollo, CSIC-Universidad Pablo de Olavide-Junta de Andalucía, Seville, Spain. [6] The Jackson Laboratory, Bar Harbor, ME, USA. [7] Laboratory for Bioinformatics Research, RIKEN BDR, Wako City, Saitama, Japan. [8] Department of Molecular Genetics and Microbiology, Genetics Institute, University of Florida, Gainesville, FL, USA. [9] These authors contributed equally: Jonas Malkmus, Laurène Ramos Martins. ✉email: aimee.zuniga@unibas.ch; rolf.zeller@unibas.ch

Precise spatio-temporal gene regulation is a defining feature of embryonic development[1,2]. Gene expression is orchestrated by *cis*-regulatory modules (CRMs) functioning as transcriptional enhancers or repressors that are most often embedded in large genomic landscapes[3], and mutations in CRMs are a major cause of congenital malformations and disease[2,4]. Genetic analysis of the genomic landscapes of developmental regulator genes indicated that functional redundancy among enhancers (or shadow enhancers)[5,6] is one of the mechanisms underlying the *cis*-regulatory robustness of gene expression and developmental processes[7,8]. The mouse limb bud is a model of choice to study the molecular interactions underlying the robustness of signalling and gene regulatory networks. One paradigm is the self-regulatory SHH/GREM1/AER-FGF feedback signalling system that controls limb bud outgrowth and patterning[9]. We previously established that pathway inter-connectivity underlies these self-regulatory properties that balance BMP and SHH activities by feedback regulation via the BMP antagonist GREMLIN1 (GREM1). These feedback interactions provide limb bud outgrowth and patterning with systems robustness[10,11]. A paramount feature of this self-regulatory signalling system is transcriptional regulation of *Grem1* by the different signalling pathways. *Grem1* functions as a key node and alterations in its expression impact both feedback regulation and the progression of limb bud outgrowth and patterning[10,12–14]. Whether the *cis*-regulatory interactions that control *Grem1* expression could provide an additional level of robustness to limb bud development is an intriguing possibility that remained to be explored. In this study, we identify the multiple CRMs that control *Grem1* expression in mouse limb buds. In-depth genetic and molecular analysis does not reveal clearly discernible redundancy. Instead, we uncover a *Grem1* core enhancer network embedded in a ~190 kb topologically associating domain (TAD) that regulates transcript levels in an additive manner while interactions among enhancers provide *cis*-regulatory robustness to the spatial regulation of *Grem1* expression during mouse limb bud development. The enhancer activities of two deeply conserved CRMs from different tetrapods and basal fishes display significant spatial differences that match the observed species-specific spatial variations in *Grem1* expression during limb bud development in different tetrapods. This evolutionary analysis provides insights into the *cis*-regulatory and spatial changes in *Grem1* expression during the fin-to-limb transition and prefigure the evolutionary diversification of the distal limb skeletal pattern (this study and refs. [15–17]).

## Results

### CRMs in the *Grem1* TAD integrate signalling inputs into gene expression.

The mouse *Grem1* and *Formin1* (*Fmn1*) genes share the same *cis*-regulatory landscape[13] encompassing the ~240 kb *Fmn1* and ~190 kb *Grem1* TAD (Fig. 1a, b, Supplementary Fig. 1)[18,19]. Genetic inactivation of *Grem1* disrupts limb skeletal patterning, which results in fusion of ulna and radius and three rudimentary digits[20,21]. In contrast, disruption of *Fmn1* does not alter limb development[13], but deletion of an ~180 kb genomic region overlapping the *Grem1-Fmn1* TAD disrupts *Grem1* expression in limb buds (*delCis*, Fig. 1b, Table 1)[13,22]. Within the *delCis* region, eight CRMs were identified by open chromatin (ATAC-seq) and active enhancer mark (histone H3K27 acetylation) profiling in mouse forelimb buds (Fig. 1b, Supplementary Figs. 2, 3), some of which overlap conserved non-coding regions identified previously (CRM2 to CRM4, Table 1)[13,22–24]. During the onset of *Grem1* expression and limb bud outgrowth, only CRM2,-3 and CRM7,-8 are part of accessible chromatin regions (E9.75) while the others are accessible by E10.5 (Fig. 1b,

Table 1 Nomenclature of the *cis*-regulatory regions in the mouse *Grem1* genomic landscape.

| CRM/EC nomenclature | Previous name | Reference |
|---|---|---|
| delCis | FmnΔ10-24 | Zuniga et al.[13] |
| EC1 | GCR | Zuniga et al.[13] |
| EC2 | | This study |
| CRM1 | | This study |
| CRM2 | GRS1 | Zuniga et al.[22] |
| CRM2: CE region | HMCO1 | Zuniga et al.[22] |
| CRM2: ME region | | This study |
| CRM3 | HMCO2 | Zuniga et al.[22] |
| CRM4 | GRE1 | Li et al.[23] |
| CRM5 | | This study |
| CRM6 | | This study |
| CRM7 | | This study |
| CRM8 | | This study |
| CRM9 | | This study |
| CRM10 - CRM13 | | This study |

Listed are the unified novel nomenclature and previous names of the *cis*-regulatory regions identified.

Supplementary Fig. 2). The potential CRM enhancer activities were assessed using *LacZ* reporter assays in transgenic mouse embryos (Fig. 1c, Supplementary Fig. 2). This identified CRM2-5 and CRM7 as active enhancers that recapitulate spatial aspects of *Grem1* expression (Fig. 1c). In contrast, the CRM6 activity is low and variable (Fig. 1c, Supplementary Fig. 2) and no *LacZ* activity is detected for CRM8 and CRM9 (Fig. 1c). CRM1 is located outside the *delCis* region and its activity does not overlap *Grem1* expression (Supplementary Fig. 2). This analysis also identified additional CRM enhancers located in the *Fmn1* TAD, two of which are active in the apical ectodermal ridge (AER, Supplementary Fig. 1) as expected from *Fmn1* expression[13]. This analysis establishes that the CRM enhancers with *Grem1*-like activities are located in the *Grem1* TAD (Fig. 1a–c).

*Grem1* expression in limb buds is regulated by transacting inputs that include BMP/SMAD4, SHH/GLI, and HOX13 transcription factor complexes[10,11,24–26]. ChIP-seq analysis identified the CRMs that integrate these *trans*-regulatory inputs into *Grem1* expression (Fig. 1d, Supplementary Fig. 3). For SMAD4, a single ChIP-seq peak is detected in CRM2 during forelimb bud formation (E.9.5–9.75, Fig. 1d) as expected from BMP4-mediated activation of *Grem1* expression[10,25]. During limb bud outgrowth, HOXA13/D13 and GLI3 ChIP-seq peaks are detected in all CRMs of the *Grem1* but not *Fmn1* TAD (E11.5, Fig. 1d, Supplementary Fig. 1; HOX13 datasets from ref. [27]). This shows that the SHH pathway and HOX13 impact *Grem1* *cis*-regulation globally rather than via specific CRMs, which points to potential *cis*-regulatory redundancy (Fig. 1c, d, Table in Supplementary Fig. 2).

### Multiple enhancers orchestrate *Grem1* expression in limb buds.

Previous genetic analysis showed that several larger genomic deletions overlap a genomic region that could be required for *Grem1* expression (termed GCR)[13], which encompasses the CRM2 to CRM4 enhancers (Fig. 1b–d, Table 1)[22,23]. Together with *Grem1* intra-TAD interactions (Fig. 1a) this led us to assign these three CRMs to one putative enhancer cluster, called EC1, while the more closely spaced CRM5 to CRM8 regions were assigned to a second cluster, EC2 (Fig. 1a, d). Both putative enhancer clusters were deleted using CRISPR/Cas9-mediated genome editing (Fig. 2). Chromatin conformation capture (4C-seq) establishes that the loss of interactions with the *Grem1* promoter is limited to the deleted regions in mutant forelimb

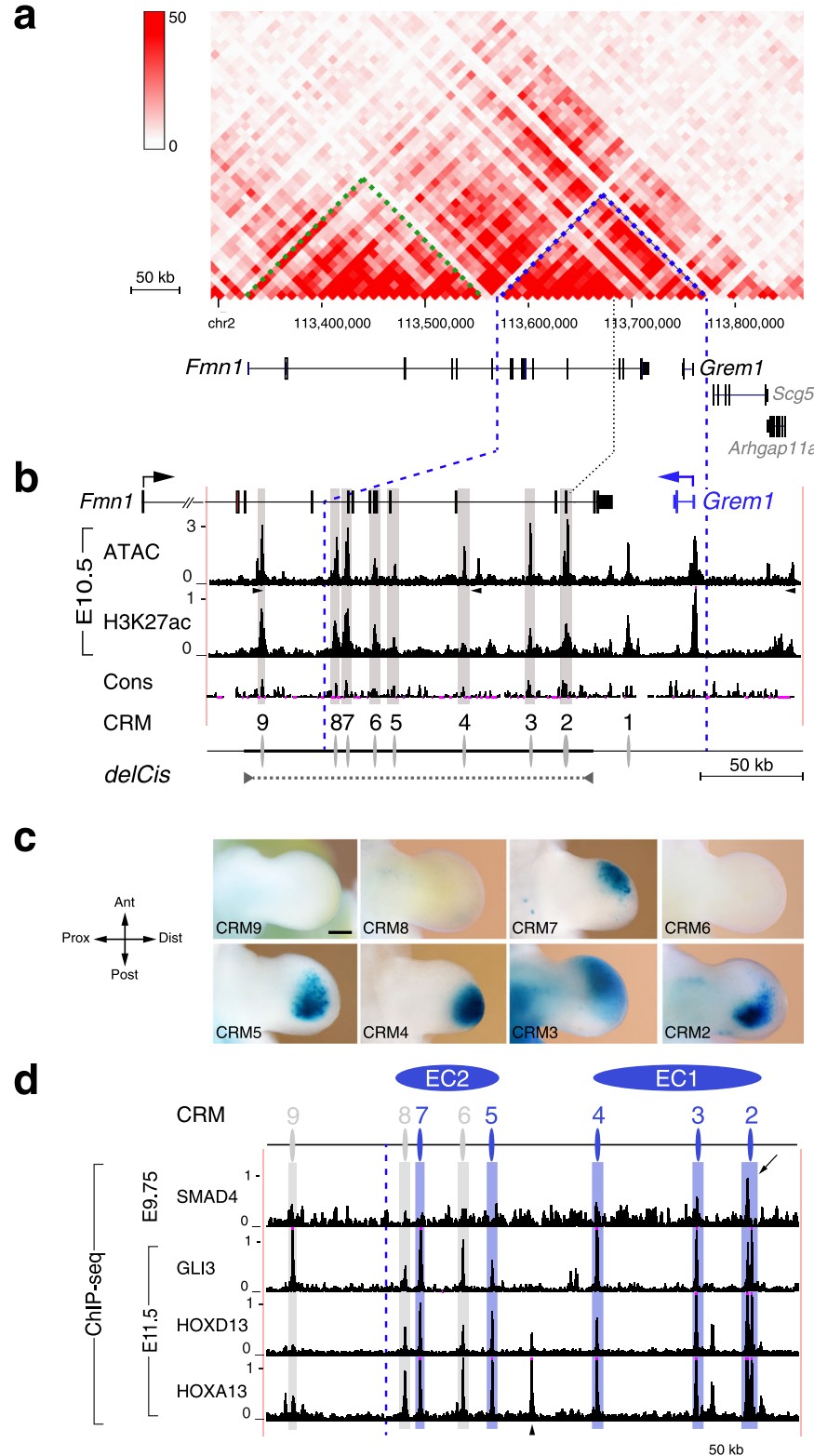

buds (Fig. 2a) and shows that the reduction or loss of *Grem1* expression (Fig. 2b–e) is due to the enhancer deletions rather than global alterations of chromatin structure (Fig. 2a). This contrasts with the widespread alterations in *delCis* homozygous forelimb buds (Supplementary Fig. 4), which are a possible consequence of deleting the 3′ boundary of the *Grem1* TAD[13,18]. Specific deletion of both enhancer clusters ($EC1^{\Delta/\Delta}EC2^{\Delta/\Delta}$) disrupts *Grem1* expression and results in a loss-of-function digit phenotype, but

does not globally disrupt the chromatin interactions with the promoter (Fig. 2c). This shows that EC1 and EC2 regions encode all CRMs essential for limb bud mesenchymal *Grem1* expression (Fig. 2a–c). Deleting either *EC2* or *EC1* reduces *Grem1* transcript levels during forelimb bud outgrowth (E11.0) by ~50% in both cases (panel RT-qPCR, Fig. 2d, e). However, comparative RNA in situ hybridisation analysis reveals spatio-temporal *Grem1* expression differences during limb bud development (Fig. 2b–e).

**Fig. 1 Multiple CRMs in the *Grem1* TAD are enhancers interacting with key transcription complexes. a** Hi-C metadata from mouse embryonic fibroblasts[19] show the chromatin interactions in the *Grem1-Fmn1* TAD on mouse chromosome 2. The colour intensity scale shows the contact frequencies. The *Grem1* TAD: ~190 kb, indicated by blue dashed lines, *Fmn1* TAD: ~240 kb, indicated by green dashed lines. *Arhgap11a* and *Scg5* are part of the genomic region but not located within the *Grem1* TAD. **b** Enlargement of the *Grem1* TAD (vertical blue dashed lines) and the *delCis* region required for *Grem1* expression in limb buds (indicated by a horizontal black dashed line). The directions of transcription are indicated by arrows. The ATAC-seq peaks (open chromatin) and histone H3K27 acetylation ChIP-seq peaks (H3K27ac; active enhancers) detected in forelimb buds at E10.5 identify all candidate CRMs located distal to the *Grem1* coding region. *n* = 2 independent biological replicates were analysed for the ATAC-seq and the H3K27ac ChIP-seq. The peak calling function of MACS2 identified the significantly enriched peaks present in both replicates of the ATAC-seq and the H3K27ac ChIP-seq. The genomic regions enriched in both ATAC-seq and H3K27ac ChIP-seq peaks correspond to candidate CRMs that are numbered in 3′ direction starting with CRM1 (Table 1) and CTCF sites that are indicated by black arrowheads. **c** *LacZ* reporter assays in transgenic founder embryos, representing independent transgene insertion events, establish robust enhancer activities for CRM2 (*n* = 7/11 expressors), CRM3 (*n* = 5/8), CRM4 (*n* = 6/8) CRM5 (*n* = 7/13) and CRM7 (*n* = 3/6) in forelimb buds (E11.0–E11.5). CRM6 displays mostly no or rarely variable activity (*n* = 2/14 expressors), while CRM8 (*n* = 0/5) and CRM9 (*n* = 0/4) are not active in limb buds. The transgenic founder embryos that express *LacZ* in forelimb buds are indicated as the fraction of all embryos with *LacZ* expression in limb and non-limb tissues. Scale bar: 250 μm. Ant: anterior, Dist: distal, Post: posterior, Prox: proximal. **d** ChIP-seq analysis identifies the interaction of SMAD4 chromatin complexes with CRM2 in the *Cis* region during the onset of forelimb development (E9.5–9.75) and the GLI3, HOXD13 and HOXA13 chromatin complexes during outgrowth (E11.5). *n* = 2 independent biological replicates were analysed for all ChIP-seq experiments and the peak calling function of MACS2 identified the significantly enriched peaks in both replicates. These peaks overlap the CRMs identified with exception of one HOXA13 ChIP-seq peak that is located in a conserved region of non-accessible and non-H3K27ac marked chromatin (indicated by an arrowhead)[22]. The only called SMAD4 ChIP-seq peak within the *Grem1* TAD is indicated by an arrow. CRM enhancers are indicated in blue, CRMs without *LacZ* activity in grey. EC1 and EC2: enhancer cluster 1/2. The inputs for the H3K27ac, SMAD4 and GLI3 ChIP-seq analyses (panels **b**, **d**) are shown in Supplementary Fig. 3.

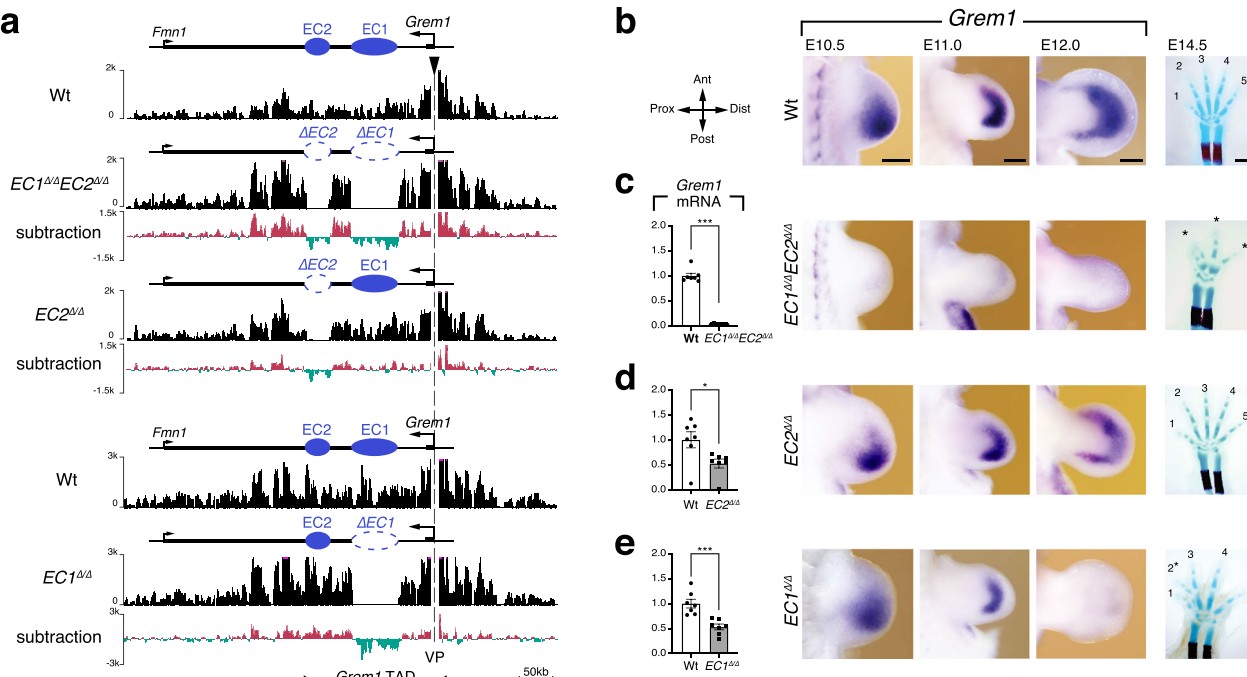

**Fig. 2 Two enhancer clusters in the *Grem1* TAD control the spatio-temporal dynamics of limb bud mesenchymal *Grem1* expression. a** Chromatin conformation capture (4C) using the *Grem1* promoter as viewpoint (VP, also indicated by a black arrowhead) to reveal its interactions with the *Grem1-Fmn1* landscape. The 4C profiles of forelimb buds lacking both EC1 and EC2 or EC2 and EC1 alone were compared to their respective wild-type controls (upper Wt: control for $EC1^{\Delta/\Delta}EC2^{\Delta/\Delta}$ and $EC2^{\Delta/\Delta}$, lower Wt: control for $EC1^{\Delta/\Delta}$ forelimb buds). Subtraction after normalization reveals that the deletions do not affect the interactions of the *Grem1* promoter with the remainder of the *Grem1-Fmn1* TAD (subtraction, green: reduction or loss of interactions, red: gain of interactions). The position of *Grem1* TAD is indicated at the bottom of the panel. **b–e** Left panels (except **b**): RT-qPCR was used to determine the relative *Grem1* transcript levels in wild-type and homozygous mutant forelimb buds (*n* = 7 independent biological replicates at E11.0, 40–42 somites per genotype). Bars represent mean values +/− SEM. *P*-values were determined using the two-tailed Mann–Whitney test: ***$p$ = 0.000583 (panels **c**, **e**), *$p$ = 0.017483 (panel **d**). Middle panels: In situ hybridisation shows the spatio-temporal *Grem1* distribution in wild-type (Wt) and mutant forelimb buds at three developmental stages (*n* = 4 embryos analysed per genotype and stage from different litters and in minimally two independent experiments). E10.5: 35–37 somites; E11.0: 40–42 somites, E12.0: staged by developmental time). Ant: anterior, Dist: distal, Post: posterior, Prox: proximal. Scale bars: 250 μm. Right panels: limb skeletons at ~E14.5 (blue: cartilage, red: ossification centres in radius and ulna). Digits are shown from anterior (digit 1) to posterior (digit 5). Only three rudimentary digits form in $EC1^{\Delta/\Delta}EC2^{\Delta/\Delta}$ forelimb buds (indicated by asterisks, *n* = 5). In contrast, pentadactyly is perfectly maintained in $EC2^{\Delta/\Delta}$ forelimbs (*n* = 3), while 64% of all $EC1^{\Delta/\Delta}$ (*n* = 9/14) forelimb skeletons display variable fusions of digits 2 and 3 (asterisk). Scale bar: 1 mm. RT-qPCR source data are provided in a Source data file.

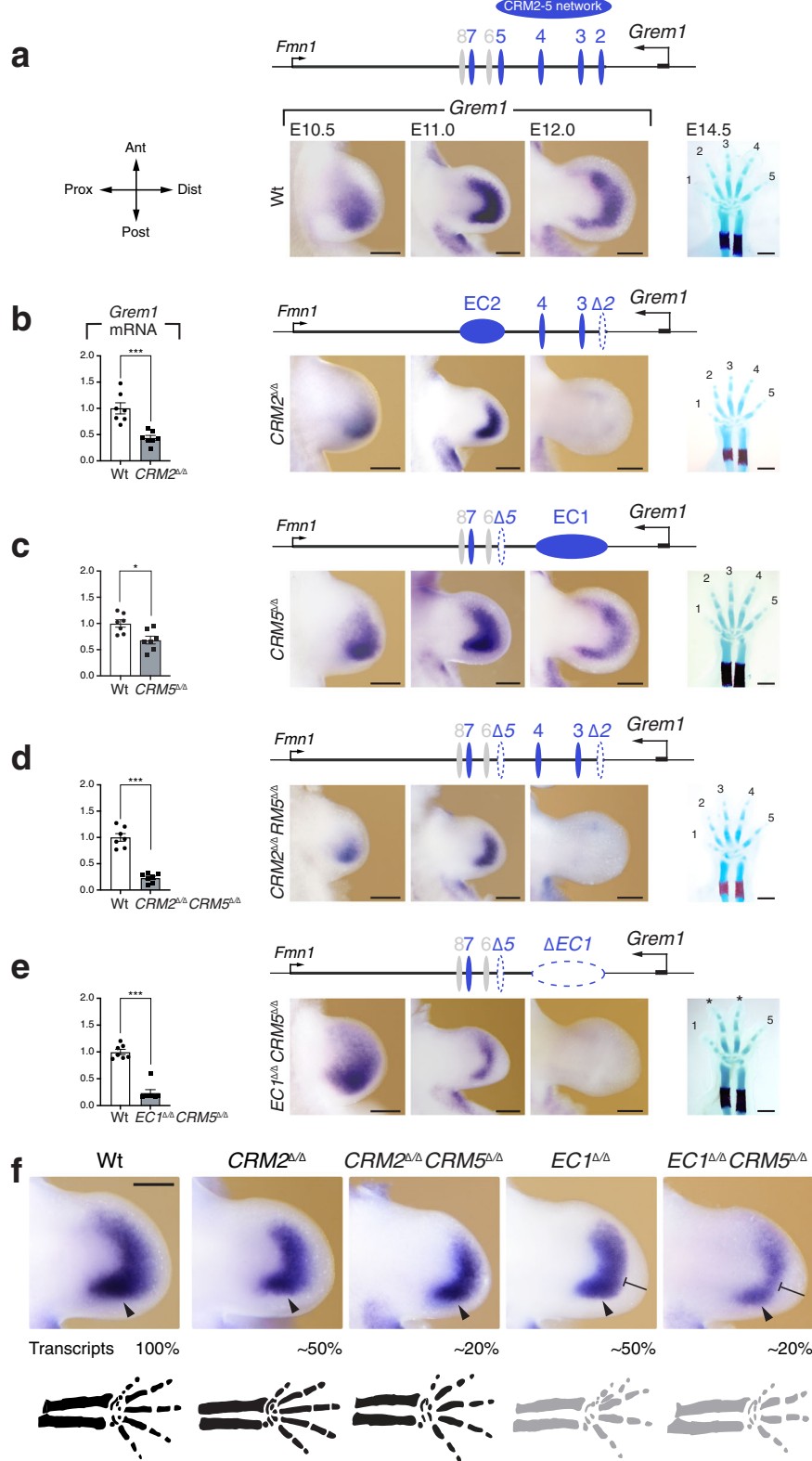

In $EC1^{\Delta/\Delta}$ and $EC2^{\Delta/\Delta}$ forelimb buds Grem1 expression is activated normally, while no activation is detected in $EC1^{\Delta/\Delta}EC2^{\Delta/\Delta}$ forelimb buds (Supplementary Fig. 4). In $EC2^{\Delta/\Delta}$ forelimb buds, the dynamic Grem1 expression pattern is similar to wild-type limb buds, i.e. expands distal-anterior into a crescent-shaped domain that retains its posterior bias (E10.5, E11.0, Fig. 2b, d; Supplementary Fig. 4)[22]. In contrast, this posterior bias is reduced

such that the Grem1 domain appears smaller and more symmetrical in $EC1^{\Delta/\Delta}$ forelimb buds (E11.0, Fig. 2e) and expression terminates precociously during mutant handplate (autopod) development (E11.0–E12.0, Fig. 2e). The comparative analysis of $EC1^{\Delta/\Delta}$ and $EC2^{\Delta/\Delta}$ forelimb buds indicates that the ~50% reduced Grem1 levels have no effect on digit patterning (Fig. 2d), while the spatio-temporal changes in $EC1$-deficient forelimb buds

**Fig. 3 Interactions among CRMs provide the spatially dynamic *Grem1* expression and pentadactyly with *cis*-regulatory robustness. a** Left panel: indication of the relevant limb bud axes. Ant: anterior, Dist: distal, Post: posterior, Prox: proximal. **b–e** Left panels: RT-qPCR was used to determine the relative *Grem1* transcript levels by comparing wild-type and $CRM2^{\Delta/\Delta}$ (**b**), $CRM5^{\Delta/\Delta}$ (**c**), $CRM2^{\Delta/\Delta}CRM5^{\Delta/\Delta}$ (**d**) and $EC1^{\Delta/\Delta}CRM5^{\Delta/\Delta}$ forelimb buds (**e**; $n = 7$ independent biological replicates at E11.0, 40–42 somites for all genotype). Bars represent mean values $+/-$ SEM. *P*-values were determined using the two-tailed Mann–Whitney test: ****p* = 0.000583 (panels **b**, **d**, **e**) and **p* = 0.011072 (panel **c**). Middle panels: spatial *Grem1* expression in wild-type (**a**) and the different single and compound mutant forelimb buds (**b–e**) at the developmental stages indicated ($n = 4$ embryos analysed per genotype and stage from different litters and in minimally two independent experiments). Scale bars: 250 μm. Right panels: limb skeletons at ~E14.5 (blue: cartilage, red: ossification centre in radius and ulna). Digits are shown from anterior (digit 1) to posterior (digit 5). Pentadactyly is maintained in $CRM2^{\Delta/\Delta}$ ($n = 4$ embryos), $CRM5^{\Delta/\Delta}$ ($n = 8$) and $CRM2^{\Delta/\Delta}CRM5^{\Delta/\Delta}$ ($n = 3$) forelimb skeletons. In contrast, all $EC1^{\Delta/\Delta}CRM5^{\Delta/\Delta}$ ($n = 8$) forelimb skeletons are tetradactyl with symmetrical middle digits of equal length (indicated by asterisks). Scale bar: 1 mm. **f** Direct comparison of the *Grem1* expression domain in forelimb buds of the most relevant *CRM* loss-of-function alleles (E11.0, 40–42 somites). These forelimb buds belong to the same group of biological replicates as the ones shown in Figs. 2 and 3. Arrowheads indicate the posterior domain. Bar-ended lines point to the stunted crescent domain. The forelimb buds are representative of the spatial distributions in the respective genotype. The relative *Grem1* transcript levels in comparison to the wild-type (set at 100%, see before) and schematics of the distal limb skeletons are shown below. Skeletons in black: pentadactyly maintained, grey: pentadactyly altered/disrupted. RT-qPCR source data are provided in a Source data file.

are the likely cause of the partial and variable digit 2/3 fusions (Fig. 2e, see also Fig. 3f).

**The CRM2-5 enhancer network provides *cis*-regulatory robustness.** To gain insight into the CRM functions and interactions, additional *Grem1* alleles lacking one or several enhancers were generated (Fig. 3 and Supplementary Fig. 5). Deletion of CRM2 ($CRM2^{\Delta/\Delta}$, Fig. 3b) reduces *Grem1* transcript levels to a similar extent as the *EC1* deficiency (~50% at E11.0, panel RT-qPCR). As for the *EC1* deficiency (Fig. 2e), *Grem1* expression is reduced and terminates prematurely in $CRM2^{\Delta/\Delta}$ forelimb buds while the posterior bias in *Grem1* expression and pentadactyly are maintained (Fig. 3b, see also Fig. 3f). Nevertheless, the reduction in *Grem1* transcript levels and premature termination in $CRM2^{\Delta/\Delta}$ forelimb buds are the most severe alterations resulting from the deletion of a single CRM enhancer. This reveals the essential *cis*-regulatory functions of CRM2, which is the enhancer in EC1 that is located closest to the *Grem1* gene. In contrast, deletion of either CRM3 or CRM4 (previously called *GRE1*, Table 1)[23] does not significantly alter the levels and spatial distribution of *Grem1* transcripts ($CRM3^{\Delta/\Delta}$: Supplementary Fig. 5, $CRM4^{\Delta/\Delta}$: ref. [23]). Furthermore, the deletion of either CRM3 or CRM4 in context of the CRM2 deficiency ($CRM2^{\Delta/\Delta}CRM3^{\Delta/\Delta}$, $CRM2^{\Delta/\Delta}CRM4^{\Delta/\Delta}$, Supplementary Fig. 5) does not reproduce the spatial *Grem1* expression changes observed in *EC1*-deficient forelimb buds (Fig. 2e) and pentadactyly is maintained (Supplementary Fig. 5). The similar spatial activities of CRM2 and CRM5 (Fig. 1c) led us to analyse *Grem1* alleles lacking CRM5 and compound mutant alleles (Fig. 3c–e). In $CRM5^{\Delta/\Delta}$ forelimb buds, *Grem1* transcript levels are lowered by ~30%, but as for the *EC2* deficiency (Fig. 2d) no spatial changes are detected and pentadactyly is maintained (Fig. 3c). In $CRM2^{\Delta/\Delta}CRM5^{\Delta/\Delta}$ forelimb buds, *Grem1* transcript levels are reduced in an additive manner by ~80% (Fig. 3d). In contrast, the spatial expression remains similar to the *Grem1* domain in $CRM2^{\Delta/\Delta}$ forelimb buds and pentadactyly is maintained in spite of the ~80% reduction in transcripts (Fig. 3d, compare to Fig. 3b). In $EC1^{\Delta/\Delta}CRM5^{\Delta/\Delta}$ forelimb buds, *Grem1* transcript levels are also reduced by ~80% and striking spatial changes in *Grem1* distribution are observed in contrast to $CRM2^{\Delta/\Delta}CRM5^{\Delta/\Delta}$ forelimb buds (Fig. 3e, compare to in Fig. 3d). During early limb bud outgrowth (E10.5), *Grem1* expression is anteriorly expanded in $EC1^{\Delta/\Delta}CRM5^{\Delta/\Delta}$ forelimb buds, but subsequently restricts to a narrow symmetrical domain (E11.0, Fig. 3e). These spatial changes are paralleled by tetradactyly with partial loss of identities in $EC1^{\Delta/\Delta}CRM5^{\Delta/\Delta}$ forelimbs (Fig. 3e).

Genetic analysis of the *Grem1 cis*-regulatory landscape (Figs. 2 and 3) establishes that four of the seven CRMs, namely CRM2 to

CRM5 are part of the core enhancer network that regulates *Grem1* distribution in mouse limb buds (Fig. 3f). In both $CRM2^{\Delta/\Delta}$ and $CRM2^{\Delta/\Delta}CRM5^{\Delta/\Delta}$ forelimb buds, the posteriorly biased and crescent-shaped *Grem1* expression domain are maintained in spite of the stepwise reduction in *Grem1* transcript levels (Fig. 3f). Deletion of either CRM2 to CRM4 ($EC1^{\Delta/\Delta}$) or CRM2 to CRM5 ($EC1^{\Delta/\Delta}CRM5^{\Delta/\Delta}$) does not alter transcript levels further, but it either weakens or disrupts the spatial regulation of *Grem1* expression and digit development (Fig. 3f). The spatial alterations in *Grem1* expression result either in reduction ($EC1^{\Delta/\Delta}$) or loss of the posterior bias ($EC1^{\Delta/\Delta}CRM5^{\Delta/\Delta}$, arrowheads) and a distally stunted crescent domain in both types of mutant forelimb buds (bar-ended line, Fig. 3f). The significantly reduced and symmetrical *Grem1* domain in $EC1^{\Delta/\Delta}CRM5^{\Delta/\Delta}$ forelimb buds results in tetradactyly with symmetrical middle digits (Fig. 3e, f), which bears remarkable resemblance with the spatial *Grem1* expression in bovine and pig limb buds and the morphological alterations of the distal limb skeleton in these *Artiodactyl* species[15,16,28].

**High plasticity of ancient *Grem1* enhancers during tetrapod evolution.** Previous analysis of limb bud mesenchymal *Grem1* expression in different tetrapods provided evidence that the spatio-temporal plasticity in *Grem1* expression correlates well with evolutionary diversification of the distal limb skeleton[15,16,29]. This prompted us to investigate how the *cis*-regulatory complexity underlying *Grem1* expression in limb buds might have arisen and diverged. To this aim, the *Grem1* TAD sequences from species representing different vertebrate clades were aligned to identify evolutionary conserved non-coding regions. In addition to *Mammalia* and *Sauropsida*, this comparison also included basal fishes: Coelacanth (*Latimeria chalumnae*), a lobe-finned fish that diverged from the lineage leading to tetrapods ~410 million years (myr) ago[30] and two cartilaginous fishes (*Chondrichthyes*), elephant shark (*Callorhinchus milii*) and bamboo shark (*Chiloscyllium punctatum*) that diverged from other jawed vertebrates ~450 myr ago[31,32]. This analysis using either the mouse (Fig. 4a), chicken or bamboo shark as reference genome (Supplementary Fig. 6) revealed the deep overall conservation of the *Grem1-Fmn1* genomic landscape and the ancient nature of the CRM2 and CRM5 enhancers, and CRM8. This analysis also shows that four of the five enhancers (CRM2/3 in EC1, CRM5/7 in EC2) are present in *Sauropsida* which diverged from *Mammalia* ~330 myr ago. Furthermore, phylogenetic analysis of CRM2 and CRM5 reveals their significant sequence diversification during tetrapod evolution (Supplementary Fig. 7). As these two ancient enhancers are the key components of the CRM2-5 enhancer network in mammals (Fig. 4a), changes in their activities could provide

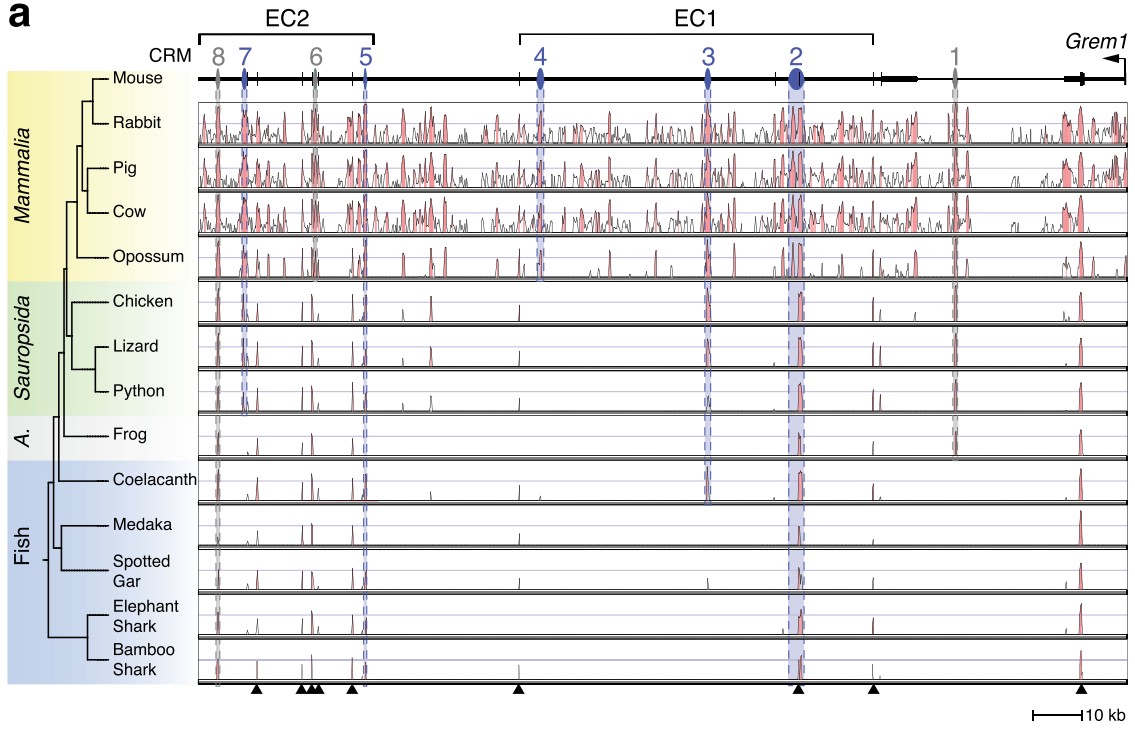

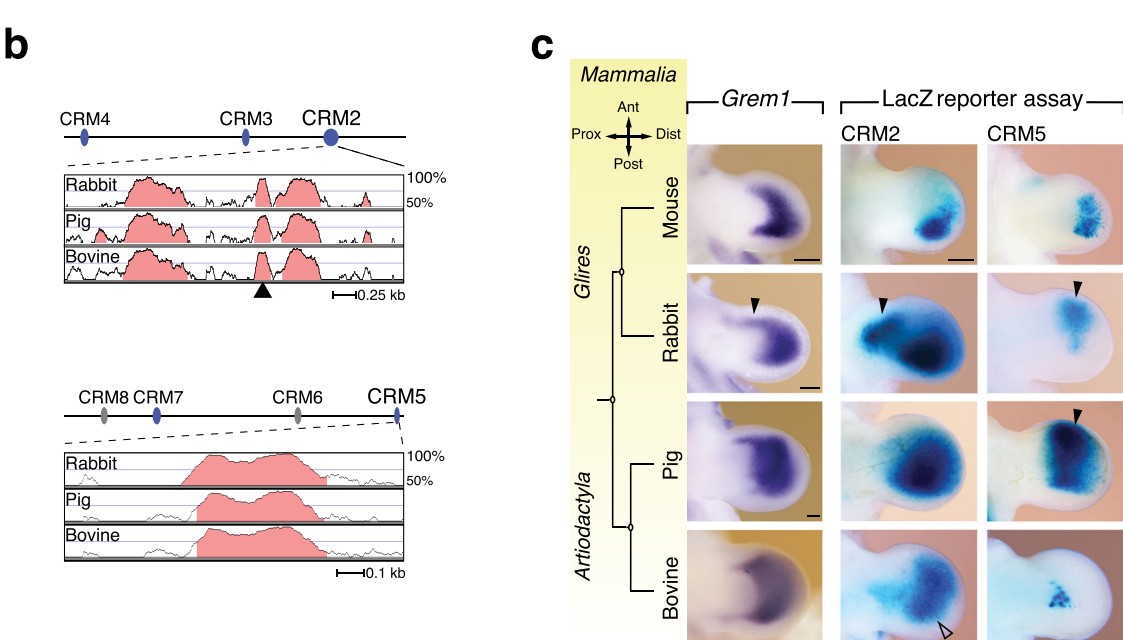

insights into the spatial plasticity underlying *Grem1* expression in mammalian limb buds (Fig. 4b, c)[15,16,28]. Therefore, the enhancer activities were compared with *Grem1* expression in limb buds of two pentadactyl (*Rodentia*: mouse, *Leporidae*: rabbit) and two even-toed artiodactyl species (*Suidae*: pig, *Bovidae*: bovine, Fig. 4b, c, Supplementary Fig. 7). In mouse and rabbit forelimb buds (Fig. 4c), *Grem1* expression is biased posteriorly, but the crescent expands further anterior-proximal in rabbit forelimb buds (arrowheads Fig. 4c, Supplementary Fig. 7). *LacZ* reporter assays show that rabbit CRM2 has the highest activity in the posterior mesenchyme, but is also active in anterior mesenchyme together with CRM5 in a pattern overlapping the *Grem1* domain in rabbit forelimb buds (arrowheads, Fig. 4c). The characteristic symmetrical *Grem1* domain in *Artiodactyl* limb buds prefigures

the paraxonic limb skeleton and digit loss (Fig. 4c, Supplementary Fig. 7)[15]. This loss of asymmetry is paralleled by anteriorly expanded activity of pig CRM5 (arrowhead), while its CRM2 orthologue retains the posterior activity bias (Fig. 4c). In contrast, the bovine CRM2 is lacking this posterior activity bias (open arrowhead) and CRM5 activity is low in transgenic mouse limb buds (Fig. 4c). This analysis shows that lineage-specific rather than common changes in CRM2 and CRM5 activities underlie the symmetrical limb bud mesenchymal *Grem1* expression in these two *Artiodactyl* species, which points to significant plasticity in CRM activities during mammalian limb skeletal diversification.

Comparison of the mammalian CRM2 orthologues reveals two highly conserved non-coding regions (Fig. 4a, b) but only one of these, termed core element (CE) is conserved from fishes to

**Fig. 4 Spatial plasticity of the ancient CRM2 and CRM5 enhancers during evolutionary diversification of mammalian limb development. a** Multiple sequence alignments using the mouse genome as reference reveals the deep evolutionary conservation of the *Grem1* TAD in jawed vertebrates (*Gnathostomata*). The CRM enhancers active in the mouse are indicated in blue and all others CRMs in grey. Regions with ≥70% conservation are shaded in light red. Black arrowheads indicate the conserved *Fmn1* exons. A: amphibians. Genomes from the following species are included. Mouse: *Mus musculus* (reference genome); rabbit: *Oryctolagus cuniculus*; pig: *Sus scrofa*; bovine: *Bos taurus*; opossum: *Monodelphis domestica*; chicken: *Gallus gallus*; lizard: *Anolis carolinensis*; python: *Python bivittatus*; frog: *Nanorana parkeri*; coelacanth: *Latimeria chalumnae*; medaka: *Oryzias latipes*; spotted gar: *Lepisosteus oculatus*; elephant shark: *Callorhinchus milii*, bamboo shark: *Chiloscyllium punctatum*. **b** Conservation plots reveal the evolutionary conservation of the relevant mammalian CRM2 and CRM5 regions in comparison to the mouse. All highly conserved regions (≥70%, shaded light red) were included in the *LacZ* reporter constructs. Black arrowhead indicates *Fmn1* exon 22 that is an integral part of CRM2 in all species. **c** Evolutionary diversification of the *Grem1* expression domains and spatial activities of CRM2 and CRM5 in pentadactyl (mouse, rabbit) and artiodactyl (pig, bovine) species. Left panels: *Grem1* expression in mouse (E11.0), rabbit (gestational day D12), pig (D23) and bovine (D34) forelimb buds ($n \geq 3$ independent embryos analysed). Middle and right panels: CRM2 and CRM5 enhancer activities from the different species as determined by *LacZ* reporter assays in transgenic mouse limb buds mouse CRM2: $n = 7/11$, CRM5 $n = 7/13$ (see Fig. 1c); rabbit CRM2 $n = 10/11$ and CRM5 $n = 4/6$ with highly variable activities; pig CRM2 $n = 3/3$ and CRM5 $n = 5/5$, bovine CRM2 $n = 4/4$ and CRM5 $n = 5/5$. Black arrowheads indicate the anterior expansion of *Grem1* expression/enhancer activities compared to the mouse. Bovine CRM2: open arrowhead indicates the loss in posterior activity bias. The transgenic founder embryos that express *LacZ* in forelimb buds are indicated as the fraction of all embryos with *LacZ* expression in limb and non-limb tissues. Ant: anterior, Dist: distal, Post: posterior, Prox: proximal. Scale bars: 250 µm.

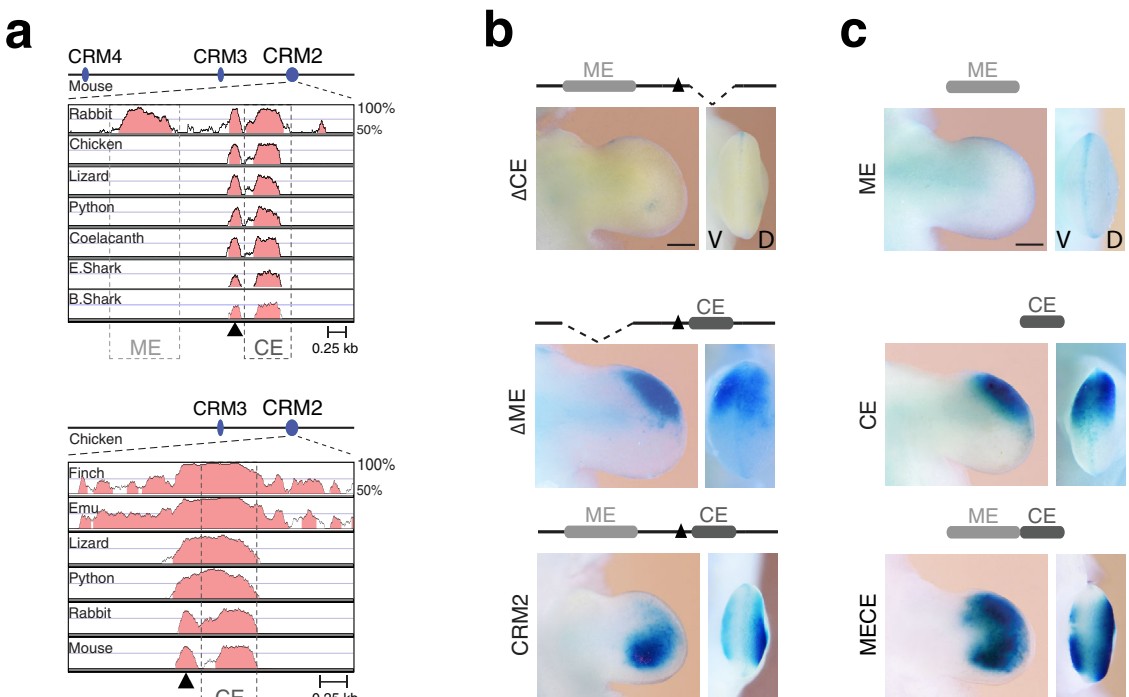

**Fig. 5 The deeply conserved CE region is essential for CRM2 activity but posterior activity depends on interaction with the ME region. a** Conservation plot analysis using the mouse (upper panel) and chicken (lower panel) CRM2 regions reveals the deep evolutionary conservation of CE region and *Fmn1* exon 22 (black arrowhead). In contrast, the upstream ME region that is conserved among mammalian species, is not detected in *Sauropsida* and basal fishes. Note the extensive conservation of the CRM2 region among different bird species. Regions with ≥70% conservation are shaded in light red. **b** Deletion analysis of the mouse CRM2 enhancer activity. Upper panel: deletion of the deeply conserved CE region (ΔCE) abolishes *LacZ* reporter activity ($n = 6/7$, proximal activity $n = 1/7$). Note that a few posterior cells with enhancer activity have been observed ($n = 2/7$). Middle panel: deletion of the mammalian-specific ME region (ΔME) restricts *LacZ* reporter activity to the anterior-distal limb bud mesenchyme ($n = 5/6$) in comparison to the intact CRM2 enhancer (lower panel, $n = 9/11$). **c** On its own, the ME region has no activity (upper panel, $n = 0/4$), while the CE region is active in the anterior-distal mesenchyme (middle panel, $n = 4/5$, activity throughout the limb bud $n = 1/5$). A *LacZ* reporter construct encoding both the ME and CE regions restores posterior activity but also retains anterior expression (lower panel, $n = 7/7$). The dorso-ventral bias in CRM2 activity together with the restriction from the sub-ectodermal region as observed for *Grem1* expression[22] is also partially restored ($n = 7/7$). The transgenic founder embryos that express *LacZ* in forelimb buds are indicated as the fraction of all embryos with *LacZ* expression in limb and non-limb tissues. Scale bars: 250 µm.

mammals (Fig. 5a), while the other is a mammalian-specific element (ME; Fig. 5a: mouse and chicken reference genomes; Supplementary Fig. 8: bamboo shark reference genome). Among bird CRM2 orthologues, significant parts of the non-coding regions are conserved in addition to the CE region (lower panel,

Fig. 5a). However, in chicken limb buds these regions are neither part of active chromatin nor enriched in HOX13 chromatin complexes (Supplementary Fig. 8). This indicates that there is no direct functional correspondence to the ME region in the chicken genome. Functional mapping of the mouse CRM2 using *LacZ*

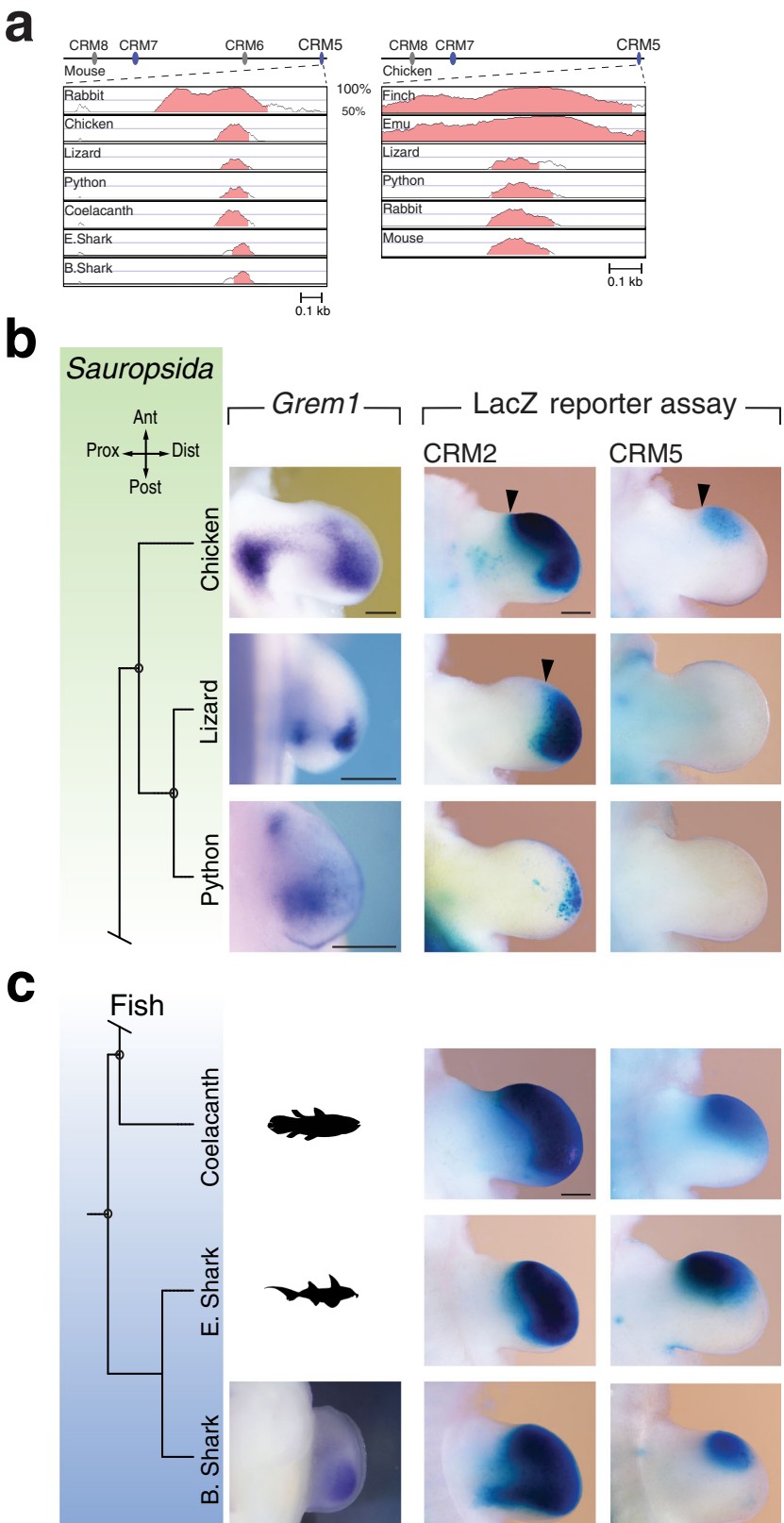

reporter constructs shows that deletion of the CE region abolishes enhancer activity (ΔCE, upper panel Fig. 5b), while the ME deletion disrupts the posterior CRM2 activity (ΔME, middle panel, compare to CRM2 lower panel, Fig. 5b). The ME and CE regions on their own have no or only anterior activity, respectively, while a construct encoding both regions is active in the posterior and anterior-distal limb bud mesenchyme (Fig. 5c and Supplementary Fig. 8). As the CE region and adjacent *Fmn1* exon 22 are deeply conserved and part of one accessible chromatin region in mouse limb buds (Fig. 1b), this entire CE22 region was also assessed, which revealed its activity in the autopod primordia (Supplementary Fig. 8). Taken together,

**Fig. 6 Reporter assays in transgenic mouse embryos reveal the limb enhancer activities of CRM2 and CRM5 from *Sauropsida* and basal fishes. a** Conservation plot analysis using the mouse (left panel) and chicken genome (right panel) as reference genomes reveals the reduced conservation of CRM5 in non-mammalian species. For using bamboo shark as reference genome see Supplementary Fig. 9. The highest conserved CRM5 regions (≥70%) are shaded in light red. **b** Left panels: *Grem1* expression in chicken wing buds (n = 4 embryos at two stages, shown is stage HH24-25) and *Anolis* lizard forelimb buds (n = 4 embryos at stage 6) at stages similar to mouse forelimb buds at E11.0. For python embryos, vestigial hindlimb buds prior to developmental arrest are shown (n = 4 embryos analysed at stage 2). Middle and right panels show representative CRM2 and CRM5 *LacZ* reporter patterns in independent transgenic mouse limb buds for the orthologues from chicken (CRM2 n = 4/4 and CRM5 n = 6/9 expressors), lizard (CRM2 n = 4/4 and CRM5 n = 1/7 expressors) and python (CRM2 n = 8/10 and CRM5 n = 0/5 expressors). Black arrowheads indicate the anterior expansion/shift of CRM2 and CRM5 enhancer activities (in comparison to their mouse counterparts, see e.g. Fig. 4c). Ant: anterior, Dist: distal, Post: posterior, Prox: proximal. **c** *LacZ* reporter assays in independent transgenic founder embryos reveal the strong expression of the conserved coelacanth, elephant- and bamboo shark CRM2 and CRM5 core enhancer regions in the distal autopod territory of transgenic mouse limb buds. Transgenic founder embryos with limb bud activities: coelacanth: CRM2 n = 9/10, CRM5 n = 7/8; elephant shark: CRM2 n = 6/6, CMR5 n = 4/4; bamboo shark: CRM2 n = 5/5, CRM5 n = 4/4 of all embryos with *LacZ* expression in limb and non-limb tissues. The *Grem1* expression in the posterior mesenchyme of paired pectoral fin buds of bamboo shark embryos is shown in the lower left panel (n = 2 embryos at slightly different stages, see Supplementary Fig. 9e). Scale bars: 250 μm. The coelacanth and elephant shark schemes are from the open access PhyloPic website (http://phylopic.org); coelacanth: Public Domain Mark 1.0; elephant shark: the Creative Commons Attribution-ShareAlike 3.0 Unported license). The elephant shark scheme was created by Tony Ayling and vectorized by Milton Tan.

the CRM2 enhancer consists of the essential and deeply conserved CE and exon 22 region and additional non-coding elements. The latter are conserved only in specific tetrapod classes such as *Mammalia* (ME region, upper panel Fig. 5a) and *Aves* as evidenced by extended conservation of CRM2 from different bird species (chicken, finch, emu, lower panel, Fig. 5a).

**Ancient fish and *Sauropsid* enhancers are active in the mouse autopod.** Similar to CRM2 (Fig. 5a), the evolutionary conservation of CRM5 between *Mammalia*, *Sauropsida* and basal fishes is restricted to a core region (Fig. 6a, Supplementary Fig. 9). During chicken wing bud (3 digits) and pentadactyl lizard forelimb bud development[33], *Grem1* is also expressed in a proximal domain in addition to the posterior-distal domain (Fig. 6b, Supplementary Fig. 9). This spatial pattern is similar to *Grem1* expression in limb buds of other bird species[29]. The chicken and lizard CRM2 enhancers are active throughout the distal mouse limb bud mesenchyme, while chicken CRM5 activity is restricted to the distal-anterior mesenchyme (arrowheads, Fig. 6b). In contrast, the lizard CRM5 is not active in transgenic mouse limb buds (Fig. 6b). Python embryos lack forelimb buds, but *Grem1* expression is activated in their rudimentary hindlimb buds (Fig. 6b, Supplementary Fig. 9). Python CRM2 activity is reduced to a small distal domain while no CRM5 activity is detected (Fig. 6b), in line with the widespread enhancer degeneration that accompanied limb loss in snakes[33–35]. Similar to mammals (Fig. 4c), the differences in *Grem1* expression in *Sauropsid* species (Fig. 6b) are due to the evolutionary diversification of CRMs that impacted their enhancer activities (Supplementary Fig. 7).

Rather unexpectedly, the CRM2 and CRM5 from lobed finned and cartilaginous fishes display strong activities in the developing mouse autopod and distal-anterior mesenchyme, respectively (Fig. 6c). In particular, the robust CRM2 enhancer activity of these basal fishes in transgenic mouse limb buds is strikingly similar to their *Sauropsid* orthologues (chicken, lizard in Fig. 6b). However, this contrasts with the posteriorly restricted *Grem1* expression in paired fin buds of bamboo shark embryos (lower panel, Fig. 6c and Supplementary Fig. 9). This discrepancy is a likely consequence of the fish CRM2 and CRM5 responding to the pathways regulating *Grem1* expression in the developing mouse autopod, namely the SHH/GLI signalling pathway and HOX transcription regulators which have also been implicated in the fin-to-limb transition[36–38]. Therefore, we assessed the effects of mutating their respective binding sites in the conserved CE region of bamboo shark CRM2 (Fig. 7a). Mutation of all Gli and Hox13 binding sites separately or combined mutation of the three

shared Hox/Gli binding sites disrupts the robust bamboo shark CRM2 enhancer (Fig. 7b), which results in variable low (Gli binding sites, Fig. 7c) or no activity (Hox13 and Gli/Hox13 binding sites, Fig. 7d, e). This analysis indicates that the evolutionary ancient *Grem1* enhancers from fishes indeed respond to the inputs that regulate *Grem1* expression in the mouse autopod.

## Discussion

Analysis of developmental regulator genes embedded in large genomic landscapes showed that redundancy among enhancers can act as regulatory buffer against variation, which manifests itself by the absence of overt phenotypes following inactivation of individual enhancers[7,8]. Our analysis also pointed to functional redundancy among CRMs, but molecular analysis indicates that the CRM2-5 core enhancer network governs limb bud mesenchymal *Grem1* expression by two distinct *cis*-regulatory principles: (1) *Grem1* transcript levels are regulated in an additive manner similar to what has been shown for the multiple enhancers controlling *Ihh* levels in mouse forelimb buds and other tissues[39]. However, altering this additive regulation of *Grem1* transcript levels has no significant effect on limb bud skeletal development as an ~80% reduction is not sufficient to disrupt pentadactyly, which indicates that *Grem1* levels are not critical to normal digit development. (2) More importantly, the genetic analysis points to synergistic interactions among CRM enhancers in regulating the spatial *Grem1* expression kinetics. In fact, individual *CRM* deletions have no discernible effects on spatial regulation with exception of CRM2, whose genetic inactivation causes only minor spatial changes but premature termination of *Grem1* expression (this study) and CRM4, whose inactivation causes subtle spatial alterations[23]. Compound mutants lacking two of the four enhancers do not alter the overall shape of the *Grem1* expression domain (this study). One possible explanation for the lack of significant spatial alterations in these mutants is functional redundancy, which has been proposed to underlie the *cis*-regulation of *Ptch1*, *Gli3* and *Fgf8* by multiple enhancers and/or so-called shadow enhancers during embryonic and/or limb bud development[8,40,41]. In contrast to mutants lacking one or two CRMs, spatial changes in the *Grem1* domain are observed in *EC1*-deficient and much more strikingly in *EC1CRM5*-deficient limb buds. The striking spatial differences between these two compound *CRM* loss-of-function mutants and all others, including the *EC2* (*CRM5-8*) deletion indicate that *cis*-regulatory robustness is disrupted when a threshold reduction in *CRM2-5* activities is reached. Therefore, it is possible that not

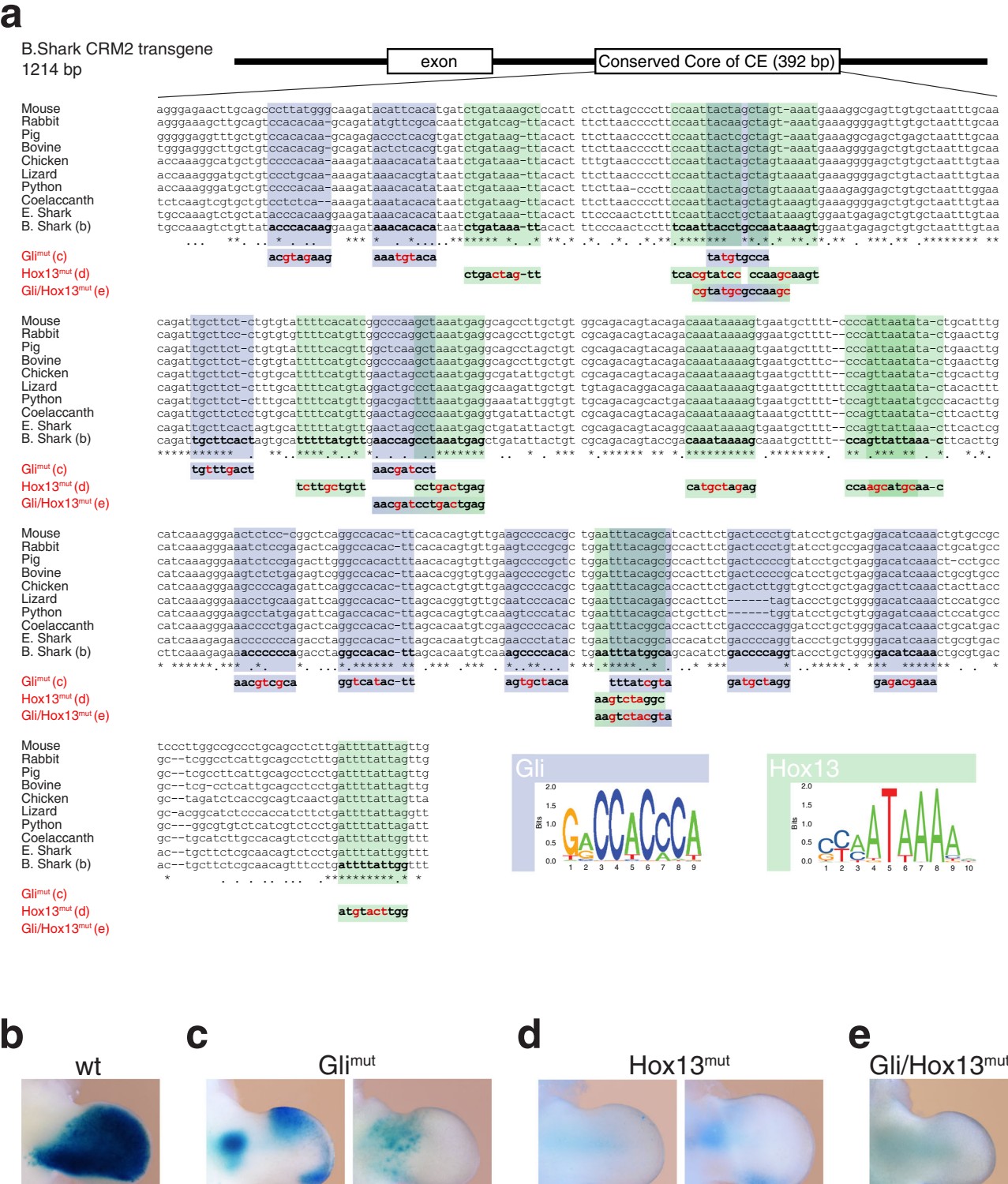

only redundant, but also interdependent and/or cooperative interactions among these CRM enhancers[2,41] govern the spatial regulation of *Grem1* expression in mouse limb buds. Cooperativity among CRM2-5 core enhancers could provide the *cis*-regulation of *Grem1* expression and pentadactyly with the high-level robustness to variation as observed by loss-of-function analysis (this study). The spatio-temporal expression of *5'HoxD* genes is also regulated by interactions involving several enhancer clusters[42], but it is unclear whether these enhancers function in a

manner similar to the *Grem1* CRM2-5 core enhancer network. Furthermore, all functionally relevant CRM enhancers are able to integrate inputs from HOX13 transcription factors and SHH/GLI signal transduction into *Grem1* *cis*-regulation, which likely strengthens robustness of the self-regulatory limb bud signalling system[10]. This signalling system and *cis*-regulatory robustness provide a likely explanation for the extreme scarcity of human congenital limb malformations linked to the *Grem1* locus[21,43]. This contrasts with the high prevalence of human congenital limb

**Fig. 7 Mutagenesis of Gli and Hox13 binding sites in the CE region disrupts the bamboo shark CRM2 activity. a** The position of the CE region in the 1214 bp bamboo shark CRM2 transgene construct is shown schematically. The Gli and Hox13 consensus sequences used for binding site identification are shown below the alignment of the CE core region. Within this CE core region, an increased number of highly conserved Gli and Hox13 binding sites are identified by multiple sequence alignment (asterisks indicate the positions of 100% base pair conservation). Therefore, the binding sites for either all Gli (11 binding sites) or Hox13 (10 binding sites), or three regions with overlapping Hox13/Gli binding sites were mutated (all binding sites indicated in bold). The nucleotide changes are indicated in red. **b-e** The resulting bamboo shark CRM2 *LacZ* reporter constructs were analysed in forelimb buds of mouse transgenic founder embryos at ~E11.0. **b** Analysis of the wild-type bamboo shark CRM2 reveals its robust and strong activity throughout the limb bud mesenchyme ($n = 7/7$, see also lower panel in Fig. 6c). **c** Mutation of all Gli binding sites (Gli[mut]) in the CE region disrupts the bamboo shark CRM2 activity in the developing autopod. Variable activities are still detected in the posterior- and anterior-distal limb bud mesenchyme (left panel, $n = 5/8$). In other forelimb buds, no activity is detected in the autopod primordia (right panel, $n = 3/8$). **d** Mutation of all Hox13 binding sites (Hox[mut]) disrupts the bamboo shark CRM2 enhancer activity in the autopod (left panel, $n = 10/10$). However, half of the founder embryos display variable activity in the proximal forelimb bud mesenchyme (right panel, $n = 5/10$). **e** Mutation of the three overlapping Gli and Hox13 binding sites (panel **a**) completely disrupts the CRM2 enhancer activity in forelimb buds of transgenic founder embryos (Gli/Hox[mut], $n = 13/15$; two embryos retain activity in forelimb buds). The transgenic founder embryos that express *LacZ* in forelimb buds are indicated as the fraction of all embryos with *LacZ* expression in limb bud and non-limb bud tissues. Scale bar (panels **b-e**): 250 μm.

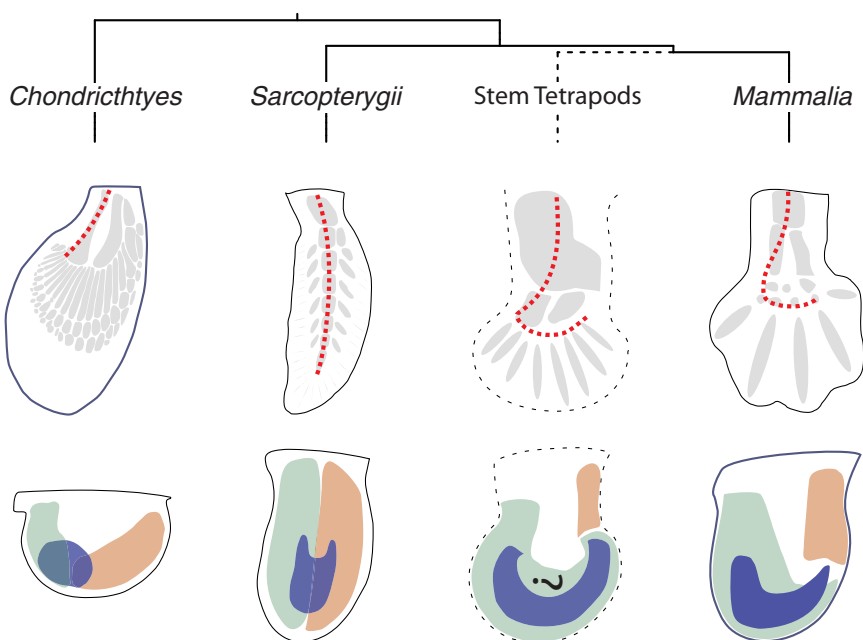

**Fig. 8 Fish and tetrapod *Grem1* expression patterns recapitulate molecular and morphological hallmarks of the fin-to-limb transition.** Upper panels: schematics of the endochondral skeletons (shaded grey) with the appendage (metapterygial) axis indicated by a red dotted line. Lower panels: schematized spatial expression of *Grem1* (blue), posterior genes (green: e.g. *5'Hoxd* and *Hand2* genes) and anterior genes (orange: e.g. *Alx4* and *Pax9* genes). The expression of posterior genes expands distal-anterior and anterior genes remain more proximally restricted during tetrapod limb bud development. In *Chondrichthyes* (bamboo shark) *Grem1* is expressed in the posterior fin bud overlapping the boundary of posterior and anterior genes. In *Sarcopterygii* (lungfish) this boundary follows the main appendage axis and *Grem1* expression is shifted to the central and distal mesenchyme. In extinct stem tetrapods (*Acanthostega*)[70] with a polydactylous autopod and distal-anteriorly bent appendage axis, the hypothetical gene expression patterns were extrapolated from polydactylous mouse limbs such as *Gli3*-deficient mice. It is likely that *Grem1* expression extended through the entire autopod as is observed for the activities of the CRM2 enhancer from different basal fishes in transgenic mouse limb buds. In pentadactyl mammals (*Mus Musculus*), *Grem1* expression is activated in the posterior mesenchyme as in fishes but then expands distal-anterior during autopod development.

malformations caused by mutations affecting single enhancers such as the one that controls *Shh* expression in limb buds[44].

Our analysis points to functional hierarchy among *Grem1* enhancers with CRM2 being the most important single CRM enhancer that is necessary for both spatial and temporal control of *Grem1* expression during limb bud outgrowth and autopod development. The CRM2 enhancer is located closest to the *Grem1* coding region and maps to open chromatin from the onset to late limb bud development. ChIP-seq analysis identifies CRM2 as *Grem1* enhancer that can integrate transacting inputs from both the BMP and SHH signalling pathways that function in activating limb bud mesenchymal *Grem1* expression in a partially redundant manner[10,11,25]. These are features reminiscent of a lead enhancer and/or an enhanceosome, the latter of which has been proposed to provide a platform for cooperative assembly of the transcriptional complexes that activate gene expression[2,7]. Furthermore, the mouse CRM2 enhancer has a very distinctive structure as it encodes the deeply conserved CE and mammalian-specific ME region and possibly additional species-specific regions that control its dynamic activity in the posterior and distal limb bud mesenchyme. The progression from spatially robust *Grem1* expression and pentadactyly to variable digit fusions and loss mimics both molecular and phenotypic features of evolutionary digit reductions and loss in mammals[15,16,28]. That the CRM2-5 enhancer network provides both *cis*-regulatory robustness and evolutionary plasticity is also apparent from the

evolutionary diversification of CRM2 and CRM5 enhancer activities, which concur with the spatial *Grem1* expression differences in limb buds of tetrapod species representing different clades and classes. However, diversification of enhancer activities does not always result in the amazing spatial plasticity that we observe for limb bud mesenchymal *Grem1* expression in various tetrapod species from different clades. For example, the activities of the enhancers controlling *krox20* expression in the developing hindbrain have significantly diversified, but the *krox20* pattern in the hindbrain remained remarkably conserved from fishes to mammals[45].

How fins transitioned to limbs is a continued source of fascination. Current models indicate that the autopod evolved by expanding the posterior mesenchyme, which enabled formation of basal radials at the expense of ectodermal fin rays (Fig. 8)[46,47]. *Grem1* is expressed in the posterior fin bud mesenchyme together with *HoxD* genes and genes of the SHH/GREM1/FGF signalling system (this study)[17,48,49], which indicates that these key regulators of *Grem1* expression were present in the common ancestor of cartilaginous fishes and tetrapods and may even date back to the origin of paired appendages[17,50–52]. This is corroborated by the fact that the ancient CRM2 enhancer from bamboo shark responds to the same *trans*-acting inputs as their mammalian counterparts, namely HOX13 transcription factors and SHH/GLI-mediated signal transduction (this study)[10,37]. This reveals the ancient and conserved nature of the *trans*/*cis*-regulatory interactions that regulate *Grem1* expression in the posterior fin bud and the distal limb bud mesenchyme. It has been postulated that during the evolutionary transition from fin to limb buds, *cis*- and *trans*-regulatory alterations caused the spatial changes resulting in distal-anterior expansion of the posterior *HoxD* expression domains (positive regulators of *Grem1*) and anterior restriction of *Gli3* expression (negative regulator of *Grem1* expression, Fig. 8)[12,25,50–53]. These spatial changes such as the distal-anterior expansion of HoxD genes could have directly co-opted *Grem1* expression to the expanding distal mesenchyme during the fin-to-limb transition. This is supported by the fact that the initial posterior expression of *Grem1* expands distal-anteriorly during fin bud outgrowth in lungfish which are the closest extant relatives to tetrapods (Fig. 8)[17]. This, together with the progressive rewiring of the archetype SHH/GREM1/FGF interactions into a feedback signalling system operating between mesenchyme and AER contributed to the increased distal outgrowth[49,54]. Comparative analysis of fish paired fin and tetrapod limb buds shows that the distal-anterior expansion of posterior genes underlies the distal-anterior turning of the appendage axis (broken red line, Fig. 8) and the gradual transition from fin rays to polydactylous digit radials[47,50–52,55,56]. This hypothesis is well supported by the fossil record that includes both tetrapodomorphs (transitional forms) and stem tetrapods[57,58]. In this context, it is interesting that CRM2 from basal fishes is active in the entire mouse autopod primordia (this study) and that uniform *Grem1* expression in mouse limb buds induces digit polydactyly due to prolonged proliferation of chondrogenic progenitors[12,59]. Therefore, it is tempting to speculate that the evolutionary rewiring of gene regulatory networks resulting in co-option of *Grem1* expression to the ancestral autopod contributed to its polydactylous nature in stem tetrapods.

## Methods

**Ethics statement and approval of all animal experimentation**. All animal experiments were performed in accordance with national laws and approved by the national and local regulatory authorities as mandated by law in Switzerland, Germany and France. In the USA and Japan, animal experimentation was approved by the Institutional Animal Care and Use Committees (IACUC).

Approval in Switzerland, mouse genetics and chicken embryos: Regional Commission on Animal Experimentation and the Cantonal Veterinary Office of the city of Basel. Approval in Germany, rabbit embryos: Niedersächsisches Landesamt für Verbraucherschutz, Oldenburg (LAVES); pig embryos: Regierung von Oberbayern - Sachgebiet 55.2 - Rechtsfragen Gesundheit, Verbraucherschutz und Pharmazie. Approval in France for bovine embryos: Comité Rennais d' Ethique en matière d'Expérimentation Animale. Approval in the USA for lizard and python embryos: University of Florida IACUC; mouse embryos for Gli3 ChIP-seq: The Jackson Laboratory IACUC. Approval in Japan: experiments using bamboo shark embryos were conducted in accordance with the guidelines approved by the IACUC at the RIKEN Kobe Branch. The 3R principles were implemented in all animal study designs and power calculations were performed and/or set standards for respective experimental analysis implemented to assure reproducibility. If possible, results were verified using complementary approaches and independent verification by different researchers. Due to genetic complexity, mice and embryos had to be genotyped prior to analysis with exception of the *LacZ* reporter analysis. Analysis included embryos of both sexes.

**Mouse strains**. Mice were housed in individually ventilated cages (Greenline-Tecniplast) at 22 °C, 55% humidity and a light cycle of 12:12 with 30 min sunrise and sunset. The mouse strain carrying the *Cis*$^{Δ/Δ}$ loss-of-function *Grem1* allele was generated previously as *Fmn*$^{Δ10.24}$ allele[13]. All other genetically altered mouse strains used for this study were generated de novo. Some of these were generated by CRISPR/Cas9 genome editing in mouse G4 ES cells and verified ES cell clones used for generation of aggregation chimeras which were then bred to germline. All others, including compound mutant stains were generated by microinjection or electroporation of the relevant single guide (sg) RNAs and CAS9 protein complexes into fertilized eggs by Center of Transgenic Models (CTM) at the University of Basel. The genomic coordinates of the *CRM* deletion *Grem1* alleles included in this study and sequences of the sgRNAs designed with CRISPOR (http://crispor.tefor.net/) and used for genome editing are listed in Supplementary Table 1. To ensure that compound mutant *CRM* alleles are located in *cis*, additional deletions were generated by re-engineering *CRM2*$^{Δ/Δ}$ and *EC1*$^{Δ/Δ}$ zygotes. The deletion alleles were identified by PCR and their exact breakpoints verified by Sanger sequencing (Microsynth.ch, Switzerland). As our analysis focuses on analysing developmental robustness, mice were backcrossed to outbred Swiss Albino mice (Janvier) and intercrossed to generate the relevant genotypes for analysis. Mice and embryos were genotyped by PCR using primer pairs for the deleted regions (Δ) and wild-type controls (Wt) listed in Supplementary Table 2.

**Embryo collection and staging**. Mouse embryos were collected from timed matings of mice with the appropriate genotypes and embryonic stages determined using somite numbers. Bovine and pig embryos were isolated from artificially inseminated cows and sows, and embryos were collected at the relevant orthologous stages[15,16]. Rabbit embryos were collected from pregnant females and staged according to the timepoint of mating, taking into consideration that ovulation is induced ∼8 h after mating (done with the help of B. Püschel and C. Viebahn at the Institute of Anatomy, University of Göttingen, Germany). Fertilized White Leghorn chicken eggs (Animalco, Switzerland) were incubated in a commercial egg incubator (38 °C, 55% humidity) and Hamburger-Hamilton (HH) stages determined prior to isolation of embryos. *Python regius* and *Anolis sagrei* were incubated in damp vermiculite at 31 and 27 °C, respectively, to develop to the desired embryonic stages. After determining the stage using morphological staging guides, the embryos were dissected from their extraembryonic membranes prior to analysis[33,60]. Eggs of the brown-banded bamboo shark were obtained from the Osaka Aquarium Kaiyukan and incubated in artificial seawater at 26 °C; embryos were collected and staged using morphological criteria such as fin bud shapes and eye pigmentation to identify stages 29 and 30[61]. The comparative analysis of *Grem1* expression in limb buds of different species was done using orthologous developmental stages whenever possible.

**ATAC-seq analysis**. About 75000 mouse limb bud cells (E9.75; E10.5 and E11.5) were used for ATAC-seq[62] analysis and per limb bud stage, $n ≥ 2$ biological replicates were analysed. The ATAC-datasets for mouse forelimb buds at E10.5 and E11.5 have been previously published and the datasets for all three stages have been validated as described[16]. This revealed the high correspondence with the ENCODE DNase-hypersensitivity sites for mouse limb buds (*R*-values: 0.76–0.79). The ATAC-seq tracks count both 5′-ends of each fragment in bins of size 10. These numbers are divided by the sum across all bins (library size) and the bin size, and then multiplied by 1e9 to obtain the reads per kilobase per million mapped reads (RPKM) value per bin. The previously unpublished ATAC-seq datasets for mouse forelimb buds at E9.75 and chicken wing buds (HH24) are available under the GEO accession number GSE151488.

**ChIP-seq analysis**. The HOXA13 and HOXD13 ChIP-seq datasets have been published previously and are publicly available (GEO: GSE81358)[27]. The H3K27ac ChIP-seq[63] was performed using mouse forelimb buds and the datasets including inputs are available under the GEO accession number GSE151488. The SMAD4 ChIP-seq was performed using mouse embryos at E9.5–9.75 and the GLI3

ChIP-seq from mouse limb buds at E11.5. These two datasets including inputs are available under the GEO accession number GSE151647. The SMAD4 ChIP-seq was generated using a FLAG epitope-tag inserted in-frame into the endogenous *Smad4* locus (*Smad4*[3xF] allele). The GLI3 ChIP-seq was performed using limb buds from mouse embryos homozygous for an N-terminal FLAG epitope-tag inserted into the endogenous *Gli3* locus (*Gli3*[3XF] allele)[40] and processed for ChIP as previously described[64]. Briefly, for all ChIP-seq analysis, limb buds were isolated at defined embryonic stages (E9.75 to E11.5). Tissue was then cross-linked for 10–20 min in 1% formaldehyde/PBS at room temperature, quenched with glycine (125 mM) and subsequently lysed in a hypotonic buffer. Sonication was used to shear chromatin and immunoprecipitation was performed overnight at 4 °C using mouse monoclonal anti-Flag antibody (Sigma F1804, 5 μg per sample) for the GLI3 and SMAD4 ChIP-seq analysis and the anti-histone H3 (acetyl K27) antibody (ChIP Grade Abcam ab4729, 5 μg per sample) for H3K27ac ChIP-seq analysis. The immune-complexed chromatin complexes were isolated using magnetic beads (Fisher Scientific 11202D). Beads were washed in RIPA buffer and the DNA was eluted from beads, which was followed by reverse cross-linking overnight. Purified DNA was used to prepare sequencing libraries using the next-generation library preparation kit from Takara Bio (Japan) according to manufacturer instructions and sequenced using a NextSeq instrument (Illumina). For H3K27ac ChIP-seq, the quality of the 41 bp high-quality paired-end reads was checked and aligned using QuasR, while for the SMAD4 ChIP-seq, the quality was checked using FastQC and Trim_Galore and high-quality reads were aligned using Bowtie. Reads mapped in proper pairs were filtered using SAMtools and peaks were called using MACS2. The genomic coordinates of the peaks were determined using BEDTools. For the GLI3 ChIP-seq analysis single-end reads of 76 bp were mapped to the mouse genome assembly GRCm38 (mm10) using bwa. Peaks were called relative to input controls using the MACS2 *callpeak* function with the following parameters: --Call-summits -B --trackline. For genome browser visualization, each ChIP-seq dataset was uniformly processed to generate tracks of fragment pileup per million reads using the -B --SPMR parameters within the macs2 callpeak utility of MACS2.

**4C chromatin conformation capture**. The 4C analysis was done as previously described[65] and the datasets are available under GEO accession number GSE151647. The following changes were implemented: for one biological replicate 2–4 × 10[6] cells were isolated from ~20 forelimb buds (E11.0, 40–42 somites). A suspension of single nuclei was made and crosslinked in 2% formaldehyde for 10 min at room temperature. Then the samples were digested with 400U of DpnII at 37 °C with gentle rotation (600 rpm). After 6 h the reaction was spiked with another 400U of DpnII and the digestion left overnight. After verification of complete digestion, samples were ligated using 100U of T4 ligase (HC, Promega) at 16 °C for 4 h, followed by 30 min at room temperature. For the second digestion, samples were diluted to 100 ng/μl and digested overnight with NlaIII (1 U/μg DNA) at 37 °C (600 rpm). Re-ligation was done using 200U of T4 DNA ligase at 16 °C for 4 h, followed by 30 min at room temperature. For the 4C analysis of *EC1*[Δ/Δ]*EC2*[Δ/Δ] and *EC2*[Δ/Δ] forelimb buds, the libraries were generated using a recently published two-step nested PCR approach[66], purified using AMPure beads (AMPure XP, Beckman Coulter) to remove fragments ≤150 bp and sequenced to generate 41 bp paired-end reads. For the 4C analysis of *EC1*[Δ/Δ] and *Cis*[Δ/Δ] forelimb buds, the libraries were generated by adding adapters and barcodes by PCR amplification (30 cycles, primers listed in Supplementary Table 3). After column purification (QIAquick PCR purification kit, Qiagen) the libraries were sequenced to generate single-end reads of ≥76 bp read length. To achieve overall high quality, raw sequencing reads that did not match the primer sequence were discarded from all samples. Filtered reads were aligned to the mouse reference genome (mm10) using Bowtie v2.2.9. To identify the valid restriction fragments, the mouse reference genome was in silico digested using DpnII and NlaIII. Restriction fragments that did not contain a cutting site for NlaIII or were smaller than 20 bp were filtered out. This yielded the library of valid restriction fragments used for quantitative analysis of experimental 4C-seq datasets. Read counts were computed for each valid fragment and the resulting 4C profile visualized using the UCSC genome browser. To visualize the data, bedGraph formatted files of the read counts for each fragment or a specified window of fragments were generated. 4C-seq contacts were analysed for the mouse region on chr2:113326224–113894862 that encompasses the *Grem1-Fmn1* landscape. The viewpoint, adjacent undigested fragments and fragments 10 kb up- and downstream were excluded. Finally, a range of 5 informative fragments was used to normalize the data per million reads (RPM) over a sliding window using custom scripts that are available at Zenodo (https://doi.org/10.5281/zenodo.5181231) and these continuous-valued profiles displayed in the UCSC genome browser tracks. These were then used to generate the panels for figures. The subtractions were computed by subtracting fragment reads for all positions of the locus between wild-type and mutant forelimb bud samples.

**Generation of CRM *LacZ* reporter transgenic mouse founder embryos**. The mouse CRM1-13 core regions and the mouse CRM2 deletion constructs were amplified by PCR from mouse genomic DNA. The primers for PCR amplification of

the target CRM regions were designed with Primer3 (https://bioinfo.ut.ee/primer3-0.4.0/) are listed together with the genomic coordinates in Supplementary Table 4. The rabbit, bovine, pig, chicken, python and elephant shark CRM2 and CRM5 orthologous regions were amplified by PCR from genomic DNA of their respective species. *Python regius* and *Anolis sagrei* lizard genomic DNAs were isolated by the Cohn group[60]. Elephant shark tissue stored in 100% ethanol was used to isolate genomic DNA with the Wizard® Genomic DNA Purification kit (Promega Inc). All primers used for amplification and the genomic coordinates of CRM2 and CRM5 in the different species and the mouse CRM2 analysis are listed in Supplementary Table 5. The coelacanth and lizard CRM2 and CRM5 regions were synthesized by Integrated DNA Technologies (USA). All CRM regions were inserted into the Hsp68-*LacZ* reporter plasmid using the Gibson assembly kit system (New England Biolabs). Transgenic mouse founder embryos were generated by the CTM using pronuclear injection and each founder embryo represents an independent biological replicate. Embryos were collected from several batches of injected embryos transferred into several pseudo-pregnant foster mothers. Mouse transgenic *LacZ* reporter assays were performed according to standard protocols[22]. Briefly, founder embryos were isolated in ice-cold PBS around E11.5 and fixed in 1% formaldehyde, 0.2% glutaraldehyde, 0.02% NP40, 0.01% sodium deoxycholate in PBS for 20–30 min at 4 °C. Subsequently, embryos were washed three times in 1xPBS for 5 min at room temperature. The reaction was performed in the dark at 37 °C in a solution containing 1 mg/mL X-Gal, 0.25 mM K3Fe(CN6), 0.25 mM K4Fe(CN6), 0.01% NP40 and 0.4 mM MgCl₂ to detect ß-galactosidase activity, which colours expressing cells in blue (=*LacZ* activity detection). Colour development was monitored and stopped toward the end of the exponential staining phase, which occurred within maximally 6–7 h for clearly positive embryos. Embryos that showed no *LacZ* staining were left overnight to possibly detect weak *LacZ* activity. All embryos were genotyped by PCR to detect the *LacZ* reporter transgene and gene copy numbers were determined for most of them. Overall, no severe biases in *LacZ* activity due to gene copy numbers were detected. To determine the spatial activities of CRM enhancers, only embryos with β-galactosidase activity were considered. The forelimb buds shown are representative for the *LacZ* patterns detected except were stated otherwise for variable patterns. In general, minimally three, but often many more founder embryos with forelimb bud *LacZ* activity formed the basis for assigning robust enhancer activity to particular CRMs. In contrast, if the vast majority of all expressing founder embryos (n ≥ 5) lacked *LacZ* activity in forelimb buds, then the CRM was scored as not active in mouse limb buds.

**Quantitative analysis of *Grem1* mRNA levels by RT-qPCR**. Embryonic limb buds (E11.0, 40–42 somites) were collected in ice-cold PBS, transferred to RNA-later (Sigma-Aldrich) and stored at −20 °C until further processing. Total RNA was extracted using the RNeasy Mini Kit (Qiagen, Germany). A minimum of seven biological replicates per genotype were generated. RT-qPCR analysis was done using *Grem1* specific primers listed in Supplementary Table 6[10]. The relative Cq values of the *Grem1* transcripts were normalized to the Cq values of the *RPL19* control and normalized fold transcript levels (2$^{-\Delta\Delta Cq}$) are shown as mean ± SEM. The data used for analysis are provided in the Source data file. The *p*-values were determined in Prism using a two-tailed Man–Whitney test.

**Whole-mount *Grem1* in situ hybridization in mouse embryos and other species**. The mouse, pig, bovine and chicken *Grem1* riboprobes have been used previously and a standard whole mount in situ hybridization protocol was used for all experiments[10]. Briefly, embryos were fixed in 4% paraformaldehyde (PFA) in PBS at 4 °C overnight, dehydrated into 100% methanol and stored at −20 °C until further use. Following rehydration, embryos were bleached in 6% hydrogen peroxide and then digested with 10 μg/ml proteinase K (10–15 min depending on the embryonic stage). Following prehybridization at 65 °C (≥3 h), embryos were incubated overnight at 70 °C in hybridization solution with 0.2–1 μg/ml heat-denatured antisense riboprobe to detect the transcripts of interest. The next day, embryos were extensively washed and non-hybridized riboprobe digested by 20 μg/ml RNase for 45 min at 37 °C. After additional washes and pre-blocking, the embryos were incubated overnight with anti-digoxigenin antibody (1:5000, Roche cat. no. 11093274910) at 4 °C. Following extensive washing to remove excess antibodies, the RNA-riboprobe hybrids were visualized by incubation in BM purple (Roche cat. no. 11442074001). The visualisation was stopped when the signal is strong but has not reached complete saturation. For comparative analysis of different stages of embryos of the same species visualisation was done for the same duration, for cross-species analysis visualisation times needed to be adjusted in a species-specific manner. The results of whole mount in situ hybridisation analyses are qualitative but well suited to detect spatial changes. The rabbit *Grem1* probe was generated by PCR amplification from embryonic cDNA (D11.5) and is orthologous to the mouse *Grem1* in situ probe. The rabbit *Grem1* probe was sequenced and first tested on mouse embryos, which detected the typical *Grem1* expression pattern. Similarly, the lizard *Grem1* probe was generated using lizard embryonic cDNA (stage 8). The lizard *Grem1* probe is orthologous to the mouse counterpart and was verified by sequencing and phylogenetic tree analysis. Lizard embryos were fixed in 4% PFA for one hour at room temperature. Python embryos were fixed overnight in 4% PFA at 4 °C. Whole mount in situ hybridization was performed using the *Grem1* probes from the lizard *Anolis sagrei* (accession number MT124663) and *Python regius* (accession number KX778825) as previously

described[60], with the following modifications for lizard embryos: the methanol dehydration step was skipped and embryos were treated with 10 μg/mL proteinase K in PBT for 15 min, blocked with 25% goat serum in KTBT. After hybridization and washing, the embryos were incubated with anti-digoxigenin antibody for 4 h at room temperature. Then, the embryos were washed two times for 15 min in KTBT solution, which was followed by an overnight wash and three additional 15 min washes prior to starting the procedure to detect the in situ signal.

**Mouse forelimb skeletal analysis**. For limb skeletal preparations, embryos were collected at E14.5–E14.75 and processed using standard protocols. Briefly, embryos were eviscerated and fixed in 95% ethanol overnight, stained for 24 h in 0.03% (w/v) Alcian blue, 80% ethanol, 20% glacial acetic acid and washed for 24 h in 95% ethanol. Embryos were then pre-cleared for 30 min in 1% KOH and counterstained in 0.005% (w/v) Alizarin in 1% (w/v) KOH. Finally, embryos were cleared with increasing concentrations of glycerol in 1% KOH (80, 60, 40 and 20%) and stored in 80% glycerol in water. Alcian blue staining detects cartilage and alizarin red ossified bone. At least three embryos per genotype were analysed.

**Vista conservation plot analysis of the deeply conserved *Grem1* TAD**. The *Grem1* TAD of the species used in this study were retrieved from UCSC or NCBI (Supplementary Table 7). The sequences were plotted using the VISTA browser (http://genome.lbl.gov/vista/) with default settings (Conserved Identity: 70%; Alignment program: Lagan) and the mouse *Grem1* TAD as a reference genome. The Vista conservation plots for CRM2 and CRM5 correspond approximately to the mouse genomic regions with transcription enhancing activities in *LacZ* reporter assays. This analysis shows that the conservation of both CRM2 and CRM5 is much lower in non-mammalian species in comparison to the mouse reference genome. To exclude bias due to using the mouse genome as sole reference, additional Vista conservation plots were generated using the chicken and bamboo shark genomes as reference.

**Phylogenetic tree inference and analysis of evolutionary rates**. To identify the orthologous CRM sequences in different species and infer the phylogenetic tree, we followed a strategy similar to the one published in ref. [35]. Briefly, the sequence of each CRM was extracted from the mouse reference genome (mm10) and then aligned against the genomes of interest using the modified bi-directional BLAST hit (BBH) method. A blastn search with mouse orthologue as query sequences was performed against every genome of interest and best hits with E-values < 1e−5 were collected. For every best hit, the genomic region of the blast alignment and the unaligned flanking regions were extracted from the different genomes. The extracted regions were extended by 20 nucleotides to account for indels. Finally, these sequences were used to query the mouse genome, best hits with E-values < 1e−5 were collected and the genomic location of the hits examined. If this location overlapped partially or completely with the genomic location of the corresponding mouse CRM, it was scored as orthologous region "detected" or else it was scored as "not detected".

For CRM2 and CRM5, the orthologous sequences from 29 different species (Supplementary Table 8) were aligned to generate a multiple sequence alignment (MSA) using MAFFT in the L-INS-i mode[35]. From the alignment, poorly aligned positions were discarded using Gblocks[67] in DNA mode allowing for 50% of gapped positions with a minimum block length of 10. Jalview was used for the visualization of the MSA[68]. For each CRM, the maximum likelihood phylogeny was inferred using IQ-TREE[69], which involves identification of the best fitting model of evolution and estimation of branch lengths. Trees were constructed with topologies using both unconstrained and constrained searches. For the constraint search, we used the known topology of the vertebrate species tree available from the UCSC genome browser (https://genome.ucsc.edu/cgi-bin/hgGateway). Agreement between these two topologies was evaluated using tree topology tests, which have been implemented in IQ-TREE to generate the phylogenetic trees as shown in Supplementary Fig. 7a, b. The phylogenetic trees generated for visualization of evolutionary relationships and shown in Figs. 4, 6, 8, and Supplementary Fig. 6 were generated with phyloT (https://phylot.biobyte.de/) based on NCBI taxonomy (https://www.ncbi.nlm.nih.gov/taxonomy) and visualized with iTOL (https://itol.embl.de/).

**Mutation of conserved Gli and Hox13 binding sites in the CE of bamboo shark CRM2**. Gli and Hox13 binding sites in the bamboo shark CRM2 were identified by scanning the genomic sequence using the PWMscan tool (https://ccg.epfl.ch/pwmtools/pwmscan.php#). For Gli binding sites, the position weight matrix (PWM) defined by Homer was used in combination with the limb GLI3 ChIP-seq dataset (E11.5). The Hox13 binding sites were identified using the PWM for HoxD13 (Jaspar ID MA0909.1). In both cases, a *p*-value cut-off of *p* < 0.01 was used for binding site identification. Mapping the binding sites for both transcription factors the mouse and bamboo shark CRM2 genomic sequences revealed their significant enrichment in the CE region. Therefore, the CE region was more precisely defined based on conservation criteria using the multiple sequence alignments generated (see above) and poorly aligned sequences were removed using the TrimAi tool (http://phylemon2.bioinfo.cipf.es/). Then, all binding sites in the

trimmed MSA that defines the CE core region were mapped (Fig. 7a). Next, the binding sites were mutated by introducing the nucleotide changes as shown in Fig. 7a. Then, the CE region was reanalysed to confirm that the nucleotide changes indeed disrupt the binding motifs and no de novo Gli or Hox13 binding sites are created. The mutated bamboo shark CRM2 regions were synthesized by Integrated DNA Technologies (IDT) and *LacZ* reporter constructs and transgenic founder embryos generated as described above.

**Reporting summary**. Further information on research design is available in the Nature Research Reporting Summary linked to this article.

## Data availability

The ATAC-seq, ChIP-seq and 4C-seq datasets generated for this study have been deposited in the gene expression omnibus (GEO) database. The chicken wing (HH24) and mouse forelimb bud ATAC-seq datasets (E9.75), and the mouse forelimb bud H3K27ac ChIP-seq datasets (E10.5, E11.5 and corresponding inputs) are available under the GEO accession number GSE151488. The SMAD4³ˣᶠ and GLI3³ˣᶠ ChIP-seq datasets (including inputs), and all 4C datasets are available under the GEO accession number GSE151647. The publicly available HOXA13 and HOXD13 ChIP-seq datasets can be found under the GEO accession number GSE81358. The *Anolis* sagrei *Grem1* mRNA (partial) is available under the GenBank accession number MT124663 and the *Python* regius *Grem1* mRNA (partial) under the GenBank accession number KX778825. Source data are provided with this paper.

## Code availability

The custom scripts generated for the analysis and visualisation of 4C-seq tracks can be downloaded from Zenodo using the following link: https://doi.org/10.5281/zenodo.5181231.

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

## Acknowledgements

We are grateful to A. Baur, J. Gamart, T. Oberholzer, N. Riesen, O. Romashkina and J. Stolte for support concerning different aspects of the study. V. Tschan participated in the gene expression analysis of the *CRM2* mutation as part of her master thesis under supervision of A.Z. E. Terszowska and A. Offinger are thanked for excellent mouse care, C. Viebahn and B. Püschel for providing access to rabbit embryos and B. Kessler and E. Wolff for pig embryos. P. Pelczar and the members of University of Basel Center for Transgenic Models (CTM) are thanked for generating all *LacZ* reporter embryos and most of the genome-edited single and compound mutant *Grem1* alleles. Dr. S. Kuraku and the Osaka Kaiyukan Aquarium are thanked for providing bamboo shark embryos (to K.O.). Sequencing was done at the Quantitative Genomics Facility of the University of Basel and ETH. Florian Geier from the DBM Bioinformatics Core Facility is thanked for analysing the ATAC-seq and H3K27ac ChIP-seq datasets. Calculations were performed using the Scientific Computing Center sci-CORE (http://scicore.unibas.ch/) at University of Basel. The sciCORE team is thanked for support in curation and storage of the genome-wide datasets. G. Andrey is thanked for expert advice on 4C analysis and input into the manuscript together with P. Tschopp. This research was initiated with support from the Bonus-of-Excellence SNF grant 310030B_166685 (to A.Z. and R.Z.) and then supported by the ERC advanced grant INTEGRAL ERC-2015-AdG; Project ID 695032 (to R.Z.) and the University of Basel provided core funding (to A.Z. and R.Z.). Additional funding support was provided by the National Institutes of Health grant R01 GM124251 (to K.A.P.). The research of J.L.R. is supported by MICINN grants BFU2017-

82974-P and MDM-2016-0687. K.O. is supported by the Special Postdoctoral Researcher Program of RIKEN.

## Author contributions

A.Z. and R.Z. conceived and supervised the study. Figures were prepared and the manuscript was written by J.M., A.Z. and R.Z. with input from all authors. J.M. performed the *Grem1* analysis for the $CRM5^{\Delta/\Delta}$, $CRM2^{\Delta/\Delta}CRM5^{\Delta/\Delta}$, $EC1^{\Delta/\Delta}CRM5^{\Delta/\Delta}$, $EC1^{\Delta/\Delta}EC2^{\Delta/\Delta}$ forelimb buds and the 4C-seq analysis for $EC2^{\Delta/\Delta}$ and $EC1^{\Delta/\Delta}EC2^{\Delta/\Delta}$ forelimb buds, and the comparative limb bud *Grem1* expression domain analysis. He performed all Vista conservation plot analysis, the CRM2 deletion (mouse) and mutation (bamboo shark) enhancer analysis and the CRM2 and CRM5 *LacZ* analysis for all species plus the rabbit, pig and chicken *Grem1* analysis. L.R.M. performed the mouse CRM *LacZ* analysis and comparative *Grem1* analysis of mouse forelimb buds carrying *CRM2* to *CRM4*, *EC1* and *delCis* deletions. She performed the 4C-seq in *delCis* and $EC1^{\Delta/\Delta}$ forelimb buds and the *Grem1* RT-qPCR analysis. S.J. curated all genome-wide datasets and performed the bioinformatics analyses for the SMAD4 ChIP-seq and 4C-seq datasets, constructed the phylogenetic trees and performed the in silico predictions of Hox13 binding sites. B.K. cloned the lizard *Grem1* gene and performed the expression analysis. V.P. did the limb skeletal and contributed to the *Grem1* analysis. R.S. performed H3K27ac and HOXD13 ChIP-seq analyses in mouse and chicken embryos. F.L. contributed to the python *Grem1* analysis and isolated the elephant shark DNA. A.D. analysed *Grem1* in bovine limb buds. J.L.R. generated all ATAC-seq datasets and pioneered the pig limb bud studies. K.A.P. performed and analysed the GLI3 ChIP-seq experiments. R.R. contributed to the *Grem1* RT-qPCR analysis. K.O. analysed *Grem1* expression in bamboo shark fin buds. M.C. supervised the lizard and python studies and gave input into the evolutionary part of this study.

## Competing interests

The authors declare no competing interests.
