## [Peer Review File · Nature Communications]

REVIEWER COMMENTS

Reviewer #1 (Remarks to the Author):

This manuscript by Laurène Ramos Martins et al reports an in-depth analysis of cis-regulatory modules (CRMs) involved in the spatio-temporal dynamic expression of *Gremlin1*, a key player of the signaling system at work in limb development. This study involves chromatin landscape analyses (chromatin accessibility, histone modifications, ...), CRM identification, CRM-mediated expression patterns (LacZ reporter transgenes), CRM functional analyses (engineered deletions), transcription factor occupancy (ChIP-seq) and CRM functional interactions (combined deletions), together with phenotypic characterization of limb development (bone, cartilage staining at E14.5). Overall this manuscript provides an extensive study of the Cis-acting elements governing the dynamic regulation of *Gremlin1* in the developing limb. It also provides evo-devo insights into the evolutionary plasticity of these CRMs. All this taken into consideration, this manuscript should be considered for publication.

However, the authors should elaborate more on what they consider to be novel in the enhancer interplay they elegantly dissected. To what extent what they qualify as "spatial cooperativity among enhancers" is distinguishable from well-described situations corresponding to enhancers interplay in patterning the hindbrain or specifying motoneuron and interneuron columns in the ventral horns of the developing spinal cord, for example, which also show inter-dependent activities. In the discussion section the authors cite the expression dynamics of 5'*Hox* genes in the distal limb bud as another context in which "spatial cooperativity" might take place. In other words, the conceptual leap provided by their study should be emphasized.

In their evo-devo analysis the authors indicate that (page 9) "During tetrapod evolution the cis-regulatory complexity increased such that four of the five enhancers are present in sauropodia". This however is biased by the fact that the multiple alignments used the mouse genome as reference (see Fig.4a and 4b). Performing similar analyses starting with the Medaka, zebrafish or shark genomes as reference might identify other conserved enhancers and a complex regulatory arrangement in fishes which would have been lost in the tetrapod phylum or gained in fish lineages. It would be informative to carry out or to show such alignments based on other genomes as reference. The authors apparently made such analyses since they refer to (page 9) "sub-TAD sequences (...) were aligned using (...) other lower vertebrates as reference genome" (as data not shown).

Minor points

On page 4 the authors indicate: "Within the delCis region, profiling of open and active chromatin identified nine potential CRMs ...". However, only CRM2 to 9 are located in this delCis region, thus 8 (not 9) potential CRMs lie in this region. As they appropriately mention on page 5: "CRM1 is located outside the critical delCis region...".

On page 5 the reasoning behind defining enhancer clusters EC1 and EC2 should be clarified. The authors state "The genomic arrangement and intra-TAD interactions led us to assign CRM2-4 to one enhancer cluster (...) and CRM5-8 to a second (...) (Fig.1a, 1d)", but this seems somehow arbitrary, at least with respect to what is shown on Figure 1a and 1d. What actually delineates EC1 and EC2 is not clear to the reader.

Page 11 last sentence "(...) is equivalent to the ancestral CRM2 already present in fishes" should appear as "(...) is equivalent to the ancestral CRM2 already present in fishes".

Legend to Figure 2e, right panel: an asterisk (2*) apparently indicates the fused digits 2-3 on the picture, this should be mentioned in the legend.

Legend to Figure 5b "LacZ reporter assays reveals (...)" should appear as "LacZ reporter assays reveal (...)".

Legend to Figure 5c, the authors indicate "The highly conserved CRM regions (>70%, shaded orange)

were included in the LacZ reporter (panels a, c)", it seems that they should refer to "(panels a, b)" Legend to suppl. Figure 5d, the authors refer to a "blue dashed box" which rather appears as a "black dashed box".

Reviewer #2 (Remarks to the Author):

In the current manuscript the authors examine how the complex temporal/spatial features of Gremlin1 limb bud expression is controlled by a battery of enhancers and the evolution of such control. They perform extensive profiling and transgenesis work to identify 9 gene regulatory elements that potentially exert some control on Gremlin1 expression, identifying CRM 2,3,4,5,7, categorized as EC1 (CRM2-4) and EC2 (CRM5-8). Through a series of transgene and deletion experiments, they analyze how the combination of these enhancers give rise to two phases of expression—an early posterior expression and a later crescent shaped expression. As no single or double combination deletion yield as strong of a phenotype on the early posterior expression as the EC1 deletion or the even more severe CRM5 EC1 deletion, the authors conclude that the enhancers must act cooperatively to elicit the posterior expression pattern. In the last part of the work the authors examine the sequence of several mammals and then several anamniote species and reporter gene activity when put into mice. Sequencewise, the lower vertebrate element consists of a core sequence, whereas the mammalian CRM has an additional region. Strikingly the shark element drives strong crescent expression in the mouse limb bud, while the expression of the gene is only found in the posterior in the shark bud. The authors speculate that this reflects the different morphogenesis of the limb bud among anamniotes and amniotes, where in mammals, the posterior region is known to expand heavily.

This is an extensive set of studies characterizing the Gremlin control elements that will be an excellent groundwork for many in the field, as Gremlin is a key regulator in the limb regulatory network. Some aspects, described below, would need further investigation to realise the full potential of the work and to bolster the main conclusions being taken.

One aspect of the data that makes it difficult to assess it is the qualitative nature of the spatial in situ hybridization data. The distance differences between the posterior expression domain and the end of the anterior expression domain are rather small at the resolution shown, so sometimes it is difficult to see the differences. A quantitation with respect to a posterior landmark would be an excellent way to classify the phenotypes more convincingly.

I am also concerned that the conclusions about enhancer cooperativity are preliminary and need fortification. In essence, through deletional analysis, the authors are inferring that since any combination of 2 EC1 sub-enhancers does not affect posterior expression as much as the deletion of the entire EC1 element. A similar argument is used for the even more extensive effect of CRM5 plus the EC1 deletion. It would greatly strengthen the authors' argument if they could reconstitute the cooperativity from the base enhancer in a series of synthetic transgene constructs. This would allow the authors to eliminate the possibility of contribution of any intervening or additional sequences that didn't pass their enhancer detection process, and also allow them to assess the contribution of the promoter region (the transgenic constructs could be cloned with the HSP base promoter or the Grem1 promoter and compared).

Can the authors exclude or consider that some of the differences in late Gremlin expression patterns are not a consequence of an earlier defect in some of the mutants?

The results of the evolution section are also striking. The authors infer that the anamniote elements are integrating the signals present in the mouse limb. Can they provide some supporting evidence on this? Such as response to inhibiting certain pathways?

Given the differences between the expression patterns, and the presence of an additional element in the mammalian CRM, some discussion or possibly analysis of potential repressive mechanisms would seem appropriate.

The speculation about the expression domain of the conserved shark CRM in mouse versus its normal expression in shark being related to the growth of posterior cells in mouse is interesting and plausible. It would benefit this section to craft the language for more non-specialist readers, and perhaps include a schema in the supplement.

Overall, the manuscript would be strengthened by focusing a deeper analysis on either the spatial or the evolutionary portions of the manuscript.

Reviewer #3 (Remarks to the Author):

This manuscript describes studies on the cis-regulatory structure of the mouse Gremlin locus. They identify and then determine the importance of the various CRMs on Gremlin-regulated limb development using genetic and genomic approaches, including multiple deletions. They show that these CRMs are bound by GLI3, HOXD13, HOXA13 and in one instance SMAD4. As expected, they find that multiple enhancers work cooperatively to regulate Gremlin levels. However, the relative levels of Gremlin do not seem to regulate the phenotypic readout per se. Instead, spatial expression is regulated cooperatively by select enhancers, a process they term 'spatial cooperativity'. With a solid understanding of the murine Gremlin locus, they then look at other vertebrates to see how key CRMs might regulate limb development across species. They find that lineage-specific rather than common changes in activities of two of these CRMs underlie symmetrical Gremlin expression in Artiodactyl limb buds. Finally, they show the same two enhancers are conserved down to basal fish lineages, indicating that the core CRMs regulating Gremlin limb development orchestrate an ancient pathway.

Strengths of this paper include the high quality of most of the work and the impressive combination of genetics, genomics and evolutionary biology. The Gremlin locus has long been a paradigm of developmental gene regulation and this study does a nice job of determining how multiple CRMs interact to regulate gene expression and how this might work across tetrapod species. A potential weakness of this work is that none of the results are particularly surprising or novel; indeed many bits of the findings presented here have been reported previously in various contexts using less sophisticated approaches. However, none of the previous studies have synthesized their findings to this degree, none of them have had the breadth of high quality experimental approaches and none of them have come close to the level of resolution afforded by these comprehensive genetic studies. Taken together, this study will be of interest to the field of developmental gene regulation.

Major concerns.

1. Figure 2 shows the effects of individually deleting EC1 and EC2 on Gremlin expression at E10.5, E11 and E12. It is critical to additionally show these two mutants at E9.5 because this will show if/how expression is spatially altered at the onset of Gremlin expression when expression might be acutely sensitive, rather than starting at E10.5 which is well after Gremlin expression is established and feedback pathways may have helped compensate for altered CRMs.

2. The GEO datasets indicate that the ATAC seq analyses shown in Fig. 1 is E9.75. Please indicate this stage on the figure/figure legend. In addition, depending on what the y-axis means (perhaps log-transformed in some cases but not in others?), the ATAC-peaks may have very low read counts (contrast this with Figure 1 in Paliou et al, 2019 PNAS where E10.5 limb ATAC peaks on the Y-axis go into the 100s or Fig. S5D where they vary between 5 and 100). If the ATAC-seq read counts are very low, the data should be replaced with a higher quality dataset (previously published – several are available), as this will give a misleadingly low number of ATAC-seq regions, potentially masking the identification of other potential enhancers.

Minor concerns

3. Figure 1D and Fig. S1A. Why does the y-scale for most (but not all) ChIP-seq/ATAC-seq traces range between 0 to 1 (or 0 to 3)? Could this instead be shown in some metric (or standardized scale) that is in common for all the ChIP-Seq tracks, including GLI3 (scaled to 50)? Presumably, the ChIP-seq data is fold-change? Please also shown input tracks for chips done at different times either in the main or supplemental figure.

4. The raw SMAD4 ChIP-seq dataset must be made available on GEO to support the results of this study even if there is another publication in the works. Please add this dataset to GEO. From what I can see in Figure 1, this data looks noisy – and there are areas of increased SMAD4 binding that although not significant, might be binding to CRMs 3 and 4 (as well as 2). This is worth mentioning because CRM2 is called a 'lead enhancer' based on its dual SMAD/GLI3 inputs while these dual inputs might actually be a common feature of EC1 CRMs if the SMAD inputs are misinterpreted.

5. The primary criteria for enhancers is, appropriately, ATAC accessibility and H3K27ac enrichment. H3K27ac enrichment is shown at E11.5, which is an odd stage given that most Gremlin regulation occurs before then and there is the potential for enhancer decommissioning (I note that this is also the stage that most enhancers are tested so it would only prevent the identification of possible transient early enhancers). To help address this, H3K27ac should be shown at E10.5 (published datasets are available) or earlier.

6. It is confusing to place the results from this study in the context of other CRMs previously described within the Gremlin locus. For comparative purposes, figure 1 should include the previous nomenclature/regions used in papers from the Zuniga and Vokes group (cited as refs 19 and 24). Again in Fig. 3, several of the CRMs previously known as HMC0s GRE could be cross-referenced).

7. Please add primers and conditions for genotyping mouse alleles to the supplementary tables or online methods.

Reviewer #4 (Remarks to the Author):

The manuscript by Laurène et al. comprehensively analyzes the putative Grem1 enhancers (CRMs) using mouse genetics. From the ATAC-seq and ChIP-seq results, they picked up 9 enhancer candidates and found these enhancers control Grem1 spatial expression cooperatively during limb bud development. Specifically, the EC1 region (which includes CRM2) and CRM5 has critical enhancer activity to maintain the pentadactyl state.

Interestingly, CRM2 and 5 are deeply conserved from basal fishes to mammals. The authors perform Grem1 in situ hybridization using multiple species embryos and LacZ mice transgenic enhancer assay using the orthologous sequences of mouse CRM2 and 5. The CRM2/5 spatial enhancer activity are changed during evolution. It seems this change supports the species-specific Grem1 expression pattern during limb development.

They also specifically focus on CRM2 and find this enhancer consists of two elements, core enhancer (CE) and mammalian-specific conserved element (ME). The CE element showed basal fish-like enhancer activity. This indicates that the mammalian CE is equivalent to the ancestral CRM2 already present in fish. In the end, the authors suggest these results provide insights into the fin-to-limb transition.

Major comments

1. Grem1 is one of the most critical genes for limb development. This intensive genetic analysis of CRMs is helpful to understand its regulation. The Smad4/Gli3/Hox13 ChIP-seq data supports that CRMs can integrate these trans-regulatory inputs into Grem1 expression. However, this reviewer has the impression that Figure 4-5 (in situ hybridization using various species embryos and LacZ enhancer assays) are descriptive and it is a little bit hard to understand the functional importance of Grem1 in fin-to-limb transition. The CE has enhancer activity like the basal fishes is not surprising from its deep conservation. If it's possible, authors should address the functional importance of Grem1 expression pattern changes in limb evolution to get the interest of broad readership of Nature communication.

2. The Grem1 expression in bamboo Shark is restricted to the posterior region. However, bamboo Shark CRM2 shows strong enhancer activity in the limb bud broadly. CRM5 also shows enhancer activity in the anterior region. These enhancer activities and actual gene expression are not consistent. The author should explain this discrepancy.

Is the assumption that the in situ hybridization result of stage 30 bamboo Shark (Supple Fig.6) is similar to the enhancer activity?

3. The mouse CE solely has a similar enhancer activity like bamboo Shark CRM2 is potentially interesting. Mouse limb phenotype with the deletion between CRM9 to ME (keeping CE intact) would be worth analyzing to address the functional importance of enhancer activity changes in evolution.

4. The ME itself does not have enhancer activity. However, it restricts the CE activity to the posterior/dorsal part (Fig.5d). This function of ME as an adjustment module looks important for limb evolution. Functional analysis of CE would be good for the manuscript. Such as the LacZ enhancer assay with small serial deletion of ME to identify the critical elements for its adjustment function.

Minor comments

1. Showing Grem1 locus TAD boundary in Fig.1a would be helpful.

2. The authors should explain in the figure legend why there are two different WT 4C-seq data in Fig.2a.

3. The part of Fig.5c is cut off from the paper.

4. I am not sure if citing unpublished results is fine. (Page 5: Sheth et al., unpublished results). Maybe provide more info.

Response to reviewers comments and suggestions

We would like to thank the four reviewers for their insightful comments and important suggestions. They have allowed us to revise the manuscript such that it is much improved in our opinion. The significant amount of additional data generated in response to the reviewers comments not only strengthen the conclusions, but include additional novel and unexpected findings that in our opinion significantly increase the impact of our study.

Reviewer #1

This manuscript by Laurène Ramos Martins et al reports an in-depth analysis of cis-regulatory modules (CRMs) involved in the spatio-temporal dynamic expression of Gremlin1, a key player of the signaling system at work in limb development. This study involves chromatin landscape analyses (chromatin accessibility, histone modifications, ...), CRM identification, CRM-mediated expression patterns (LacZ reporter transgenes), CRM functional analyses (engineered deletions), transcription factor occupancy (ChIP-seq) and CRM functional interactions (combined deletions), together with phenotypic characterization of limb development (bone, cartilage staining at E14.5). Overall this manuscript provides an extensive study of the Cis-acting elements governing the dynamic regulation of Gremlin1 in the developing limb. It also provides evo-devo insights into the evolutionary plasticity of these CRMs. All this taken into consideration, this manuscript should be considered for publication.

However, the authors should elaborate more on what they consider to be novel in the enhancer interplay they elegantly dissected. To what extent what they qualify as “spatial cooperativity among enhancers” is distinguishable from well-described situations corresponding to enhancers interplay in patterning the hindbrain or specifying motoneuron and interneuron columns in the ventral horns of the developing spinal cord, for example, which also show inter-dependent activities. In the discussion section the authors cite the expression dynamics of 5’Hox genes in the distal limb bud as another context in which “spatial cooperativity” might take place. In other words, the conceptual leap provided by their study should be emphasized.

Based on the suggestion by one of other reviewers we conducted a more detailed analysis of the spatial alterations caused by deletion of multiple enhancers shown in Fig. 4 (and Supplementary Fig. 6). This analysis reveals a threshold of robustness for spatial regulation by the cooperative interactions among the relevant CRM enhancers. In our opinion there has been no other study showing that enhancers regulate transcript levels in an additive manner and spatial expression in a cooperative manner that generates a threshold of robustness. Disruption this robustness that underlies the regulation of spatial expression and digit patterning results in reduction and loss of digits. We have extensively rewritten the relevant parts of the results (page 9) and the discussion section (page 12/13).

In their evo-devo analysis the authors indicate that (page 9) “During tetrapod evolution the cis-regulatory complexity increased such that four of the five enhancers are present in sauropodia”. This however is biased by the fact that the multiple alignments used the mouse genome as reference (see Fig.4a and 4b). Performing similar analyses starting with the Medaka, zebrafish or shark genomes as reference might identify other conserved enhancers and a complex regulatory arrangement in fishes which would have been lost in the tetrapod phylum or gained in fish lineages. It would be informative to carry out or to show such alignments based on other genomes as reference. The authors apparently made such

analyses since they refer to (page 9) “sub-TAD sequences (...) were aligned using (...) other lower vertebrates as reference genome” (as data not shown).

We agree with this reviewer and now include these data in Fig. 5, 6, Supplementary Fig.7 and Supplementary Fig. 8. The analysis of the entire *Grem1* TAD using three different reference genomes shows that we likely have not missed any major conserved CRM enhancers. As we did the same for the Vista plot analysis of CRM2 and CRM5 we realized that among birds the conservation of e.g. CRM2 extends beyond the deeply conserved CE region (see e.g. Page 10).

Minor points

On page 4 the authors indicate: “Within the delCis region, profiling of open and active chromatin identified nine potential CRMs ...”. However, only CRM2 to 9 are located in this delCis region, thus 8 (not 9) potential CRMs lie in this region. As they appropriately mention on page 5: “CRM1 is located outside the critical delCis region...”.

Well spotted...we got our maths wrong here

On page 5 the reasoning behind defining enhancer clusters EC1 and EC2 should be clarified. The authors state “The genomic arrangement and intra-TAD interactions led us to assign CRM2-4 to one enhancer cluster (...) and CRM5-8 to a second (...) (Fig.1a, 1d)”, but this seems somehow arbitrary, at least with respect to what is shown on Figure 1a and 1d. What actually delineates EC1 and EC2 is not clear to the reader.

One of the reasons for assigning CRM2-4 to one enhancer cluster EC1 is that this region corresponds to the previously identified GCR region. We now state the reasons for assigning the CRMs to the two clusters on page 5 (last paragraph) and include Table1 for easy comparison of the previous nomenclature to the unified and consistent CRM numbering in our study.

Page 11 last sentence “(...) is equivalent to the ancestral CRM2 already present in fishes” should appear as “(...) is equivalent to the ancestral CRM2 already present in fish”.

Has been changed in the revised version

Legend to Figure 2e, right panel: an asterisk (2) apparently indicates the fused digits 2-3 on the picture, this should be mentioned in the legend.*

Done.

Legend to Figure 5b “LacZ reporter assays reveals (...)” should appear as “LacZ reporter assays reveal (...)”.

Legend to Figure 5c, the authors indicate “The highly conserved CRM regions (>70%, shaded orange) were included in the LacZ reporter (panels a, c)”, it seems that they should refer to “(panels a, b)”

Done.

Legend to suppl. Figure 5d, the authors refer to a “blue dashed box” which rather appears as a “black dashed box”.

Done.

Reviewer #2

In the current manuscript the authors examine how the complex temporal/spatial features of Gremlin1 limb bud expression is controlled by a battery of enhancers and the evolution of such control. They perform extensive profiling and transgenesis work to identify 9 gene regulatory elements that potentially exert some control on Gremlin1 expression, identifying CRM 2,3,4,5,7, categorized as EC1 (CRM2-4) and EC2 (CRM5-8). Through a series of

transgene and deletion experiments, they analyze how the combination of these enhancers give rise to two phases of expression—an early posterior expression and a later crescent shaped expression. As no single or double combination deletion yield as strong of a phenotype on the early posterior expression as the EC1 deletion or the even more severe CRM5 EC1 deletion, the authors conclude that the enhancers must act cooperatively to elicit the posterior expression pattern. In the last part of the work the authors examine the sequence of several mammals and then several anamniote species and reporter gene activity when put into mice. Sequencewise, the lower vertebrate element consists of a core sequence, whereas the mammalian CRM has an additional region. Strikingly the shark element drives strong crescent expression in the mouse limb bud, while the expression of the gene is only found in the posterior in the shark bud. The authors speculate that this reflects the different morphogenesis of the limb bud among anamniotes and amniotes, where in mammals, the posterior region is known to expand heavily.

This is an extensive set of studies characterizing the Gremlin control elements that will be an excellent groundwork for many in the field, as Gremlin is a key regulator in the limb regulatory network. Some aspects, described below, would need further investigation to realise the full potential of the work and to bolster the main conclusions being taken.

*One aspect of the data that makes it difficult to assess it is the qualitative nature of the spatial in situ hybridization data. The distance differences between the posterior expression domain and the end of the anterior expression domain are rather small at the resolution shown, so sometimes it is difficult to see the differences. A quantitation with respect to a posterior landmark would be an excellent way to classify the phenotypes more convincingly. We have done this analysis in a blinded and randomized manner and show the results in the new Fig. 4 and Supplementary Fig. 6. The scatterplots reveal that the reduction in the posterior domain/bias together with the increase in the distal gap are the key changes that affect the spatial *Grem1* expression and underlie the disruption of pentadactyly. It allows us to conclude that the spatial enhancer cooperativity results in robustness with a threshold to perturbation as apparent from the distinct clustering of the EC1 and EC1CRM5 datapoints in Fig. 4b.*

*I am also concerned that the conclusions about enhancer cooperativity are preliminary and need fortification. In essence, through deletional analysis, the authors are inferring that since any combination of 2 EC1 sub-enhancers does not affect posterior expression as much as the deletion of the entire EC1 element. A similar argument is used for the even more extensive effect of CRM5 plus the EC1 deletion. It would greatly strengthen the authors' argument if they could reconstitute the cooperativity from the base enhancer in a series of synthetic transgene constructs. This would allow the authors to eliminate the possibility of contribution of any intervening or additional sequences that didn't pass their enhancer detection process, and also allow them to assess the contribution of the promoter region (the transgenic constructs could be cloned with the HSP base promoter or the *Grem1* promoter and compared).*

The loss-of-function analysis reveals the essential CRMs and their cooperative interactions in regulating the spatial dynamics of *Grem1* expression during limb bud outgrowth. This together with open chromatin, histone acetylation and key transcription factor binding patterns indicates that we must have identified the most relevant if not all CRM enhancers

orchestrating *Grem1* expression in limb buds. In fact, the depth of our genetic dissection of a large *Grem1* landscape is rather rare if not unique. However, we cannot exclude that intervening sequences and enhancer spacing play a role.

We have addressed the question of cooperativity for the two functionally relevant regions that are part of the CRM enhancers, namely the deeply conserved CE and the ME region that is only conserved in mammals (new Figure 6). This analysis reveals the modularity of the mouse CRM2 enhancer region but also shows the limitations with respect to studying the spatial dynamics of enhancer interactions using LacZ reporter assays.

This is likely due to the fact that the LacZ protein is too stable to monitor the spatial dynamics of *Grem1* expression. To be able to address the question raised by this reviewer in a convincing manner one would have to switch to a novel transgenic approach which relies on inserting the transgene into the H11 locus using CRISPR engineering, which will reduce founder to founder variability. In addition, one has to use a labile GFP protein such as Venus or similar that is able to “report” the spatial dynamics of *Grem1* expression. Over the last ≥ 6 months we have unfortunately not been able to reliably establish the H11 targeting system in our transgenic core facility. Furthermore, only a positive outcome of the proposed series of synthetic serial enhancer constructs would be informative, while a failure would be difficult to conclusively interpret. While we very much appreciate the suggestion from this reviewer, we consider this aspect unfortunately way beyond the current scope of our analysis.

Can the authors exclude or consider that some of the differences in late Gremlin expression patterns are not a consequence of an earlier defect in some of the mutants?

In Supplementary Figure 4, we now include an analysis of *Grem1* expression in $EC1^{\Delta\Delta}$ and $EC2^{\Delta\Delta}$ and $EC1^{\Delta\Delta}EC2^{\Delta\Delta}$ forelimb buds during onset of limb bud development, which shows that the activation of *Grem1* expression is disrupted only in the mutant limb bud lacking all CRM enhancers (page 6). Also we mention that in forelimb buds lacking CRM2, *Grem1* transcript levels are reduced from early stages onward, while in $EC1^{\Delta\Delta}CRM5^{\Delta\Delta}$ forelimb buds *Grem1* expression is expanded anteriorly by E10.5 and then restricts to a narrow and symmetrical domain (page 7/8- already discussion in the previous version of the manuscript). Therefore, the observed changes at early and later stages seem overall rather consistent with the later effects on expression and digit patterning.

The results of the evolution section are also striking. The authors infer that the amniote elements are integrating the signals present in the mouse limb. Can they provide some supporting evidence on this? Such as response to inhibiting certain pathways?

This was a difficult request to address as all our LacZ analysis is done in founder embryos. As *Grem1* expression in the autopod primordia is controlled by HOX13 transcription factors and the SHH/GLI pathway, we mapped and mutated all binding sites for the two transcription factors in the conserved CE region (new Supplementary Fig. 11). Mutation of all Gli and Hox13 binding sites separately or combined mutation of three shared Hox/Gli binding sites results in variable low or no bamboo shark enhancer activity, which establishes that the bamboo shark CRM2 is indeed regulated by the transacting factors that control mouse *Grem1* expression during limb bud outgrowth (page 12).

Given the differences between the expression patterns, and the presence of an additional

element in the mammalian CRM, some discussion or possibly analysis of potential repressive mechanisms would seem appropriate.

As also another reviewer requested a more detailed analysis of the CE and ME regions in CRM2 we have conducted a more detailed analysis of these elements, which reveals the modular structure of CRM2. This analysis also shows that both ME and CE conserved non-coding regions are required for posterior activity (new Fig. 6 and new Supplementary Fig. 9, page 10/11). Please note that the previous “CE region” construct also included *Fmn1* exon 22 (this construct is now called CE22).

The speculation about the expression domain of the conserved shark CRM in mouse versus its normal expression in shark being related to the growth of posterior cells in mouse is interesting and plausible. It would benefit this section to craft the language for more non-specialist readers, and perhaps include a schema in the supplement.

We agree and include a new Fig. 8 in the discussion section

Overall, the manuscript would be strengthened by focusing a deeper analysis on either the spatial or the evolutionary portions of the manuscript.

We hope that this reviewer agrees that the additional experimental analysis strengthen both the enhancer and evolutionary analysis and that the connection between both is relevant for a more general understanding.

Reviewer #3

This manuscript describes studies on the cis-regulatory structure of the mouse Gremlin locus. They identify and then determine the importance of the various CRMs on Gremlin-regulated limb development using genetic and genomic approaches, including multiple deletions. They show that these CRMs are bound by GLI3, HOXD13, HOXA13 and in one instance SMAD4. As expected, they find that multiple enhancers work cooperatively to regulate Gremlin levels. However, the relative levels of Gremlin do not seem to regulate the phenotypic readout per se. Instead, spatial expression is regulated cooperatively by select enhancers, a process they term ‘spatial cooperativity’. With a solid understanding of the murine Gremlin locus, they then look at other vertebrates to see how key CRMs might regulate limb development across species. They find that lineage-specific rather than common changes in activities of two of these CRMs underlie symmetrical Gremlin expression in Artiodactyl limb buds. Finally, they show the same two enhancers are conserved down to basal fish lineages, indicating that the core CRMs regulating Gremlin limb development orchestrate an ancient pathway.

Strengths of this paper include the high quality of most of the work and the impressive combination of genetics, genomics and evolutionary biology. The Gremlin locus has long been a paradigm of developmental gene regulation and this study does a nice job of determining how multiple CRMs interact to regulate gene expression and how this might work across tetrapod species. A potential weakness of this work is that none of the results are particularly surprising or novel; indeed many bits of the findings presented here have been reported previously in various contexts using less sophisticated approaches. However, none of the previous studies have synthesized their findings to this degree, none of them have had the breadth of high quality experimental approaches and none of them have come close to the level of resolution afforded by these comprehensive genetic studies. Taken together, this study will be of interest to the field of developmental gene regulation.

Major concerns.

1. *Figure 2 shows the effects of individually deleting EC1 and EC2 on Gremlin expression at E10.5, E11 and E12. It is critical to additionally show these two mutants at E9.5 because this will show if/how expression is spatially altered at the onset of Gremlin expression when expression might be acutely sensitive, rather than starting at E10.5 which is well after Gremlin expression is established and feedback pathways may have helped compensate for altered CRMs.*

Indeed this is a relevant important point. We have performed this analysis and include the results in Supplementary Fig. 4. (results page 6).

2. *The GEO datasets indicate that the ATAC seq analyses shown in Fig. 1 is E9.75. Please indicate this stage on the figure/figure legend. In addition, depending on what the y-axis means (perhaps log-transformed in some cases but not in others?), the ATAC-peaks may have very low read counts (contrast this with Figure 1 in Paliou et al, 2019 PNAS where E10.5 limb ATAC peaks on the Y-axis go into the 100s or Fig. S5D where they vary between 5 and 100). If the ATAC-seq read counts are very low, the data should be replaced with a higher quality dataset (previously published – several are available), as this will give a misleadingly low number of ATAC-seq regions, potentially masking the identification of other potential enhancers.*

As we now state in detail in the online methods section (page 34) all three ATAC-seq datasets used for this study (revised Fig. 1b, Supplementary Fig.2) were generated and their quality verified as part of a previous study, but only the E10.5 and E11.5 dataset were subsequently published (Tissières et al 2020, ref. 16). The ATAC-seq tracks are shown as standard (not log-transformed) reads per kilobase per million mapped reads (RPKM). The previously unpublished ATAC-seq datasets for mouse forelimb buds at E9.75 and chicken wing buds (HH24) are available under the GEO accession number GSE151488.

Minor concerns

3. *Figure 1D and Fig. S1A. Why does the y-scale for most (but not all) ChIP-seq/ATAC-seq traces range between 0 to 1 (or 0 to 3)? Could this instead be shown in some metric (or standardized scale) that is in common for all the ChIP-Seq tracks, including GLI3 (scaled to 50)? Presumably, the ChIP-seq data is fold-change? Please also shown input tracks for chips done at different times either in the main or supplemental figure.*

The reviewer is correct to point this out. In the initially submitted manuscript the GLI3-ChIP-seq track was showing the raw counts instead of normalized counts. We have re-analyzed the GLI3 ChIP-seq analysis and also processed all other ChIP-seq datasets such that they now all represent normalized counts (fragment pileup per million reads). In particular, we have used `-B --SPMR` parameters within the macs2 callpeak utility of MACS2. Now, the scale represents the fragment pileup per million reads for all the ChIP-seq data (Figure 1d; Supplementary Fig. 1c and Supplementary Fig. 3)

4. *The raw SMAD4 ChIP-seq dataset must be made available on GEO to support the results of this study even if there is another publication in the works. Please add this dataset to GEO. From what I can see in Figure 1, this data looks noisy – and there are areas of increased SMAD4 binding that although not significant, might be binding to CRMs 3 and 4 (as well as 2). This is worth mentioning because CRM2 is called a ‘lead enhancer’ based on its dual SMAD/GLI3 inputs while these dual inputs might actually be a common feature of EC1 CRMs if the SMAD inputs are misinterpreted.*

There was a misunderstanding here- what the “manuscript in preparation” referred to was the generation of the *Smad4*^{3xF} allele. The SMAD4 ChIP-seq dataset and its input are available under the GEO accession number GSE151647 (see Online methods page 34). The only significantly enriched SMAD4 peak at E9.75, i.e. during the onset of *Grem1* expression is located in CRM2 (Fig. 1d, input: Supplementary Fig. 3). It is also relevant to note that only CRM2 and CRM3 map to open chromatin at this early stage while CRM4 is not (Supplementary Fig. 2). The SMAD4 peak is only one of several criteria for proposing that CRM2 is a lead enhancer (Discussion, page 13 last paragraph).

5. The primary criteria for enhancers is, appropriately, ATAC accessibility and H3K27ac enrichment. H3K27ac enrichment is shown at E11.5, which is an odd stage given that most Gremlin regulation occurs before then and there is the potential for enhancer decommissioning (I note that this is also the stage that most enhancers are tested so it would only prevent the identification of possible transient early enhancers). To help address this, H3K27ac should be shown at E10.5 (published datasets are available) or earlier.

We now show the H3K27ac datasets at E10.5 (Fig. 1 and Supplementary Fig. 1) as suggested by this reviewer.

6. It is confusing to place the results from this study in the context of other CRMs previously described within the Gremlin locus. For comparative purposes, figure 1 should include the previous nomenclature/regions used in papers from the Zuniga and Vokes group (cited as refs 19 and 24). Again in Fig. 3, several of the CRMs previously known as HMCOS GRE could be cross-referenced).

We now include Table 1 (page 23) that lists the unifying CRM nomenclature together with the previously published nomenclature as suggested by this reviewer

7. Please add primers and conditions for genotyping mouse alleles to the supplementary tables or online methods.

The list of all genotyping primers is now Included in the online methods at the end of the section “Mouse strains”.

Reviewer #4

The manuscript by Laurène et al. comprehensively analyzes the putative Grem1 enhancers (CRMs) using mouse genetics. From the ATAC-seq and ChIP-seq results, they picked up 9 enhancer candidates and found these enhancers control Grem1 spatial expression cooperatively during limb bud development. Specifically, the EC1 region (which includes CRM2) and CRM5 has critical enhancer activity to maintain the pentadactyl state. Interestingly, CRM2 and 5 are deeply conserved from basal fishes to mammals. The authors perform Grem1 in situ hybridization using multiple species embryos and LacZ mice transgenic enhancer assay using the orthologous sequences of mouse CRM2 and 5. The CRM2/5 spatial enhancer activity are changed during evolution. It seems this change supports the species-specific Grem1 expression pattern during limb development. They also specifically focus on CRM2 and find this enhancer consists of two elements, core enhancer (CE) and mammalian-specific conserved element (ME). The CE element showed basal fish-like enhancer activity. This indicates that the mammalian CE is equivalent to the ancestral CRM2 already present in fish. In the end, the authors suggest these results provide insights into the fin-to-limb transition.

Major comments

1. Grem1 is one of the most critical genes for limb development. This intensive genetic

analysis of CRMs is helpful to understand its regulation. The Smad4/Gli3/Hox13 ChIP-seq data supports that CRMs can integrate these trans-regulatory inputs into Grem1 expression. However, this reviewer has the impression that Figure 4-5 (in situ hybridization using various species embryos and LacZ enhancer assays) are descriptive and it is a little bit hard to understand the functional importance of Grem1 in fin-to-limb transition. The CE has enhancer activity like the basal fishes is not surprising from its deep conservation. If it's possible, authors should address the functional importance of Grem1 expression pattern changes in limb evolution to get the interest of broad readership of Nature communication.

In revising the manuscript we have conducted comprehensive analysis of the mouse CRM2 enhancer (point 4 of this reviewer), which uncovers its modular structure consisting of the essential and deeply conserved CE and exon 22 region and additional non-coding elements that are conserved only in specific tetrapod classes. This provides insights into how CRM2 activities may have changed during evolutionary diversification of tetrapod limbs. In addition, we have established that the bamboo shark CRM2 responds to HOX and GLI3 transcriptional regulators in mouse limb buds, which is of direct relevance to the fin to limb transition (point 2 of this reviewer). Together, the inclusion of these new results significantly strengthen the evolutionary part of our study.

2. The Grem1 expression in bamboo Shark is restricted to the posterior region. However, bamboo Shark CRM2 shows strong enhancer activity in the limb bud broadly. CRM5 also shows enhancer activity in the anterior region. These enhancer activities and actual gene expression are not consistent. The author should explain this discrepancy. Is the assumption that the in situ hybridization result of stage 30 bamboo Shark (Supple Fig.6) is similar to the enhancer activity?

As part of the revisions we have been able to show that the bamboo shark CRM2 responds to HOX and GLI transcriptional regulators as mutations of their binding sites in the CE region disrupts the robust activity in the autopod primordia (new Supplementary Fig. 11, page 12). These results allow us to formulate a rather straightforward hypothesis of how Grem1 expression was directly co-opted during to the expanding mesenchyme as a consequence of the distal anterior expansion of HOX genes during the fin-to-limb transition (see revised discussion (page 14/15) and the new summarizing Fig. 8. The discrepancy between bamboo shark endogenous expression and enhancer activity in mouse limb buds can be explained by the evolutionary changes in the spatial expression of upstream regulators of *Grem1* expression such as 5'*HoxD* genes between shark fin and mouse limb buds.

3. The mouse CE solely has a similar enhancer activity like bamboo Shark CRM2 is potentially interesting. Mouse limb phenotype with the deletion between CRM9 to ME (keeping CE intact) would be worth analyzing to address the functional importance of enhancer activity changes in evolution.

Although this type of large deletion analysis would be interesting it is very problematic as causes massive alterations of the chromatin interactions with the promoter that extend well beyond the boundaries of the deletion. We have not only seen this for the delCis deletion (4C analysis in Supplementary Fig. 4a) but also another large deletion we engineered in the *Grem1* TAD. Therefore, we decided long ago to focus our genetic analysis on targeted deletions of single enhancers or clusters, which specifically disrupt the interactions of the deleted region with the promoter (Fig. 2a). Furthermore, in light of the results of the dissection of the mouse CRM2 enhancer in response to this reviewer's point 4,

the generation of a large deletion allele leaving only CE intact could disrupt *Grem1* expression or not be informative as both ME and CE are required for posterior activity (new Fig. 6). Last but not least, the generation, germline breeding and complete analysis of new *Grem1* alleles would in the best case take minimally one year.

4. The ME itself does not have enhancer activity. However, it restricts the CE activity to the posterior/dorsal part (Fig.5d). This function of ME as an adjustment module looks important for limb evolution. Functional analysis of CE would be good for the manuscript. Such as the LacZ enhancer assay with small serial deletion of ME to identify the critical elements for its adjustment function.

NOTE: It is important to realize that the “CE” construct in the originally submitted manuscript (original Fig. 5d , lowest panel) also included the adjacent deeply conserved *Fmn1* exon 22. Naming this construct “CE” would be misleading in light of the new analysis conducted for revision. Therefore, this construct is now called CE22 (new Supplementary Fig. 9d). We apologize for any confusion this may potentially cause.

Based on this reviewers suggestion, we have now conducted a more comprehensive analysis of the mouse CRM2 enhancer shown the new Figure 6 and Supplementary Fig 9c, d. This analysis includes LacZ reporter constructs carrying specific deletions in the mouse CRM2 and short constructs with the ME, CE and combined MECE regions. Most importantly, this analysis shows that both the mammalian-specific ME region and the deeply conserved CE region are required together for posterior activity (see the activity of the CEME construct, in Fig. 6c) . Together this analysis uncovers the modular structure of the CRM2 enhancer that consists of the essential and deeply conserved CE and exon 22 region and additional non-coding elements that are conserved only in specific tetrapod classes. For more details see page 10/11 of the result section and the new Figure 6 and Supplementary Fig 9c, d.

Minor comments

*1. Showing *Grem1* locus TAD boundary in Fig.1a would be helpful.*

Is now indicated by the blue dotted lines

2. The authors should explain in the figure legend why there are two different WT 4C-seq data in Fig.2a.

Because not all 4C analysis was done at the same time and using slightly different protocols, we include both respective controls- this is now explicitly stated in the figure legend: upper Wt: control for *EC1^{Δ/Δ}EC2^{Δ/Δ}* and *EC2^{Δ/Δ}*, lower Wt: control for *EC1^{Δ/Δ}* forelimb buds.

3. The part of Fig.5c is cut off from the paper.

That must have happened in the generation of the PDF- we will check into this during resubmission

4. I am not sure if citing unpublished results is fine. (Page 5: Sheth et al., unpublished results). Maybe provide more info.

In revising the manuscript and adding significantly more data, we have removed the statements on the unpublished WNT data as they are not essential for the study.

REVIEWER COMMENTS

Reviewer #1 (Remarks to the Author):

As already commented upon the first revision of this manuscript, this work corresponds to an extensive study of the Cis-acting elements governing the dynamic regulation of Gremlin1 in the developing limb. The key message of this manuscript concerns the functional interactions between enhancers, showing additive interactions to ensure the appropriate Grem1 transcript levels, and cooperative interactions to provide the spatial Grem1 expression kinetics and limb bud development robustness.

The few points which were raised upon the initial revision have been appropriately addressed by the authors. This taken into consideration, I now recommend this revised manuscript for publication in Nat Comm.

Reviewer #2 (Remarks to the Author):

In their revised manuscript the authors provide further characterization of the Grem1 enhancers, improved phenotyping of their mutants and novel evidence in support of the evolutionary conservation of the transcription factors action on the conserved enhancers.

The new detailed characterization of the CRM2 module and the mutational analysis of the bamboo shark CRM2 significantly strengthen the evolutionary part of the manuscript.

I appreciate the authors provided new quantitation provided in figure 4 and supplement figure 6 but I am still concerned that the evidence reported for the "spatial cooperativity" is still insufficient to support the statements present in the paper (e.g. "spatial alterations are the sole cause of digit fusions and loss"). The new figure 4 presents the quantifications of the entity of the spatial alterations. Unfortunately these quantifications are rather rudimentary and most importantly do not appear to be supported by statistical analysis. I am also concerned because the various limb buds in Figure 4a seem to have different sizes so it is not evident to me whether some of the differences in measurements are due to other confounding factors. Also, the other confounding possibility that the putative spatial phenotype in is due to an earlier defect has been addressed in supplementary figure 4, but unfortunately the information regarding transcript levels at stages of limb development before E11 is provided as "data not shown".

As the rest of the manuscript provides high quality molecular and genetic characterization of the Grem1 locus that is relevant to fields of limb development, gene regulation and evolution. I would suggest that the authors remove the measurements on the spatial cooperatively and ameliorate their interpretation of spatial cooperativity which currently detracts from the manuscript and put this idea in the discussion. I would suggest to remove references to spatial cooperativity including in the section title

"Cooperativity among CRM2 to CRM5 enhancers endows the spatially dynamic Grem1 expression and pentadactyly with cis-regulatory robustness "

Otherwise I am very supportive of publication of the manuscript as it has provided a highly extensive characterization of the Gremlin1 enhancers looking across evolution.

Reviewer #3 (Remarks to the Author):

This revised version of the manuscript has satisfactorily addressed all of the concerns in my review of the initial manuscript.

+++++

Additional comments on Reviewer #2's report from Reviewer #3:

In looking this over, I very much agree with reviewer #2 that the analysis justifying spatial cooperativity does not have any apparent statistical analysis associated with it. Worse, Fig 4B and Fig S6 are depicted in such a way that it looks like such an analysis has been done, and this is echoed in the description (the data was collected in a blinded fashion). There needs to be some type of statistical analysis that is performed here to determine if these groupings are significant. If the results aren't significant, they should remove the term spatial cooperativity from their manuscript. I am not as concerned with stage differences; the author's method of approximating this by measuring the maximal length of the AP axis seems reasonable, though I do think they should provide information in the figure legends / supplemental figure legends on the somite stages of each of the datapoints to maximize transparency (a mitigating factor in my opinion is that the limb bud is dynamic at these stages and it is really hard to get double mutants at exactly the right stage).

One of the challenges of this paper is that the authors are trying to advance what they say is a new paradigm of spatial cooperativity. I find this term rather unconvincing. I don't much care what they call it - it is frankly some type of spatial redundancy as has been shown with other developmental enhancers. I like the paper despite this wording because the work is rigorous, high quality science and it will be an important contribution to the field, regardless of the 'spatial cooperativity' angle.

Reviewer #4 (Remarks to the Author):

The authors have addressed all comments made by the reviewers. Supplementary Fig. 11 is compelling evidence to support the importance of Grem1 in fin-to-limb transition. I thus support the publication of this manuscript in Nature Communications.

Just one minor comment:

The authors should mention in the figure legend the meaning of orange and green area of limb illustration in Fig8.

Such as, 'posterior genes (5'Hox; Hand2, GREEN)'

Response to the comments and suggestions by the reviewers

We wish to thank all four reviewers for their evaluation of our revised manuscript and positive recommendation concerning its suitability for publication

Reviewer #1:

This reviewer has no further comments to be addressed and states:

“The few points which were raised upon the initial revision have been appropriately addressed by the authors. This taken into consideration, I now recommend this revised manuscript for publication in Nat Comm.

Thank you very much for your positive recommendation!

Reviewer #2 and reviewer#3: Comments on the new Figure 4 and Supplementary Figure S6

We respond to these comments here as both reviewer raise the same issues and we agree with them.

Reviewer #2:

In their revised manuscript the authors provide further characterization of the Grem1 enhancers, improved phenotyping of their mutants and novel evidence in support of the evolutionary conservation of the transcription factors action on the conserved enhancers.

The new detailed characterization of the CRM2 module and the mutational analysis of the bamboo shark CRM2 significantly strengthen the evolutionary part of the manuscript.

I appreciate the authors provided new quantitation provided in figure 4 and supplement figure 6 but I am still concerned that the evidence reported for the “spatial cooperativity” is still insufficient to support the statements present in the paper (e.g. “spatial alterations are the sole cause of digit fusions and loss”). The new figure 4 presents the quantifications of the entity of the spatial alterations. Unfortunately these quantifications are rather rudimentary and most importantly do not appear to be supported by statistical analysis. I am also concerned because the various limb buds in Figure 4a seem to have different sizes so it is not evident to me whether some of the differences in measurements are due to other confounding factors.

.....

As the rest of the manuscript provides high quality molecular and genetic characterization of the Grem1 locus that is relevant to fields of limb development, gene regulation and evolution. I would suggest that the authors remove the measurements on the spatial cooperativity and ameliorate their interpretation of spatial cooperativity which currently detracts from the manuscript and put this idea in the discussion. I would suggest to remove references to spatial cooperativity including in the section title “Cooperativity among CRM2 to CRM5 enhancers endows the spatially dynamic Grem1 expression and pentadactyly with cis-regulatory robustness “

Additional comments on Reviewer #2's report from Reviewer #3:

In looking this over, I very much agree with reviewer #2 that the analysis justifying spatial cooperativity does not have any apparent statistical analysis associated with it. Worse, Fig 4B and Fig S6 are depicted in such a way that it looks like such an analysis has been done, and this is echoed in the description (the data was collected in a blinded fashion). There needs to be some type of statistical analysis that is performed here to determine if these groupings are significant. If the results aren't significant, they should remove the term spatial cooperativity from their manuscript. I am not as concerned with stage differences; the author's method of approximating this by measuring the maximal length of the AP axis seems reasonable, though I do think they should provide information in the figure legends / supplemental figure legends on the somite stages of each of the datapoints to maximize transparency (a mitigating factor in my opinion is that the limb bud is dynamic at these stages and it is really hard to get double mutants at exactly the right stage).

One of the challenges of this paper is that the authors are trying to advance what they say is a new paradigm of spatial cooperativity. I find this term rather unconvincing. I don't much care what they call it - it is frankly some type of spatial redundancy as has been shown with other developmental enhancers. I like the paper despite this wording because the work is rigorous, high quality science and it will be an important contribution to the field, regardless of the 'spatial cooperativity' angle.

As requested in the course of the initial review, we performed measurements of several spatial parameters of the Grem1 expression domain in limb buds at E11.0 for the most relevant genotypes as at this stage spatial changes

are apparent. We decided to represent the data in form of descriptive scatter plots in Fig. 4 and Supplementary Fig. S6, last but not least to get critical feedback from the reviewers. We did the requested statistical analysis (one-way ANOVA) assessing significant differences for all the different measurements. This revealed statistically significant changes between key genotypes with the largest observed spatial differences (wt vs EC1 and wt vs. EC1CRM5). However, smaller differences between other genotypes were not significant due to variability between samples as also pointed out as a potential problem by reviewer 3 (“*a mitigating factor in my opinion is that the limb bud is dynamic at these stages and it is really hard to get double mutants at exactly the right stage*”). This analysis supports the observed key differences (distal gap widening and reduction of the posterior domain) when comparing wild-type with *EC1* and *EC1CRM5* mutant limb buds. However, the qualitative nature of the in situ analysis is not sufficient to arrive at firm conclusions and more importantly, statistically verify the correlation between all genotypes, spatial changes and the appearance of limb phenotypes. Therefore, we decided to remove Fig.4 and Supplementary Fig. S6 from the manuscript and alter the text as suggested by the two reviewers. We also removed any reference to spatial cooperativity as new *cis*-regulatory paradigm from the manuscript (see red marked text in the revised manuscript). As suggested by reviewer 2, we briefly discuss potential spatial cooperativity in the Discussion section (page 13) .

Note: we have decided to add a new panel (Fig.3f) to Fig. 3, which shows the spatial *Grem1* distribution in limb buds at E11.0 in the most relevant CRM deletions to allow direct comparison. The most relevant spatial changes are also described in the results section on page 8. As these limb buds are shown at higher magnification it is also in line with the request by reviewer 2 during the first review.

Reviewer #2 (additional comment):

Also, the other confounding possibility that the putative spatial phenotype in is due to an earlier defect has been addressed in supplementary figure 4, but unfortunately the information regarding transcript levels at stages of limb development before E11 is provided as “data not shown”.

This statement was indeed confusing as it refers to an RT-qPCR analysis done using *CRM2*-deficient limb buds at E10.5, which shows that transcript levels are reduced to the same extent as at E11.0 (Fig. 3b). As we show that reductions of *Grem1* transcript levels are not apparently linked to spatial phenotypes, we have removed this statement.

Otherwise I am very supportive of publication of the manuscript as it has provided a highly extensive characterization of the Gremlin1 enhancers looking across evolution.

Thank you for your insightful review of our revised manuscript and positive recommendation concerning publication.

Reviewer #3 (see also above):

This revised version of the manuscript has satisfactorily addressed all of the concerns in my review of the initial manuscript.

Thank you very much for your helpful input (see before) and positive evaluation.

Reviewer #4:

The authors have addressed all comments made by the reviewers.

Supplementary Fig. 11 is compelling evidence to support the importance of Grem1 in fin-to-limb transition.

I thus support the publication of this manuscript in Nature Communications.

Thank you very much for supporting publication by Nature Communications.

Just one minor comment:

The authors should mention in the figure legend the meaning of orange and green area of limb illustration in Fig8. Such as, 'posterior genes (5'Hox; Hand2, GREEN)

We thank this reviewer for pointing this out as this is a mistake. We have corrected this in the revised legend to what is now Fig. 7.

REVIEWERS' COMMENTS

Reviewer #3 (Remarks to the Author):

I am satisfied with the changes to this manuscript. It represents a valuable contribution to the field.